# Considering hormone-sensitive cancers as a single disease in the UK biobank reveals shared aetiology

Muktar Ahmed [1,2,3,4✉], Ville-Petteri Mäkinen [1,5], Anwar Mulugeta[1,3,4], Jisu Shin [1,6], Terry Boyle[1,4,6], Elina Hyppönen [1,3,4,7] & Sang Hong Lee [1,4,6,7✉]

Hormone-related cancers, including cancers of the breast, prostate, ovaries, uterine, and thyroid, globally contribute to the majority of cancer incidence. We hypothesize that hormone-sensitive cancers share common genetic risk factors that have rarely been investigated by previous genomic studies of site-specific cancers. Here, we show that considering hormone-sensitive cancers as a single disease in the UK Biobank reveals shared genetic aetiology. We observe that a significant proportion of variance in disease liability is explained by the genome-wide single nucleotide polymorphisms (SNPs), i.e., SNP-based heritability on the liability scale is estimated as 10.06% (SE 0.70%). Moreover, we find 55 genome-wide significant SNPs for the disease, using a genome-wide association study. Pair-wise analysis also estimates positive genetic correlations between some pairs of hormone-sensitive cancers although they are not statistically significant. Our finding suggests that heritable genetic factors may be a key driver in the mechanism of carcinogenesis shared by hormone-sensitive cancers.

[1] Australian Centre for Precision Health, University of South Australia, Adelaide, SA, Australia. [2] Department of Epidemiology, Faculty of Public Health, Jimma University Institute of Health, Jimma, Ethiopia. [3] UniSA Clinical and Health Sciences, University of South Australia, Adelaide, SA, Australia. [4] South Australian Health and Medical Research Institute, Adelaide, SA, Australia. [5] Computational Systems Biology Program, Precision Medicine Theme, South Australian Health and Medical Research Institute, Adelaide, SA, Australia. [6] UniSA Allied Health & Human Performance, University of South Australia, Adelaide, SA, Australia. [7] These authors jointly supervised this work: Elina Hyppönen, Sang Hong Lee. ✉email: muktar.ahmed@mymail.unisa.edu.au; Hong.Lee@unisa.edu.au

Cancer continues to dominate as one of the major global public health problems with increasing incidence[1] and multiple aetiologies[2]. The risk of cancer is in part modifiable, and demographic and lifestyle factors have been reported to explain some of the variability in cancer[3]. There is also a genetic component to cancer, evidenced from twin and sibling studies[4,5]. Large-scale genomic studies have also identified germline variants (single-nucleotide polymorphism [SNPs]) that are linked with the susceptibility to various types of cancer, using population samples[6,7].

Cancer is a broad term for a heterogeneous group of diseases, all sharing an uncontrolled cell growth. However, there is also evidence for shared mechanisms; for example, hormonal pathways affect the development of several types of cancer[8]. Group of cancers that share a characteristic mechanism of carcinogenesis that involves hormones, namely breast, uterine, prostate, ovary, testis, osteosarcoma, and thyroid cancers, are termed as hormone-sensitive cancers[9]. The role of common target genes and transcriptional cofactor has received a considerable attention in the biology of hormone-sensitive cancers. For example, the expression of fibroblast growth factor (FGF-2) gene is a signalling molecule with fundamental roles in tumour growth and progression associated with breast, ovarian, thyroid, prostate, and uterine cancers[10–14]. Furthermore, Michailidou et al., (2013)[15] reported that multiple genomic regions flanking common target genes, such as telomerase reverse transcriptase (TERT) and the POU domain class 5 transcription factor 1B (POU5F1B), included susceptibility loci that were common to breast, prostate and ovarian cancers, supporting the hypothesis of a common genetic aetiology among these cancer types[16–18]. Cancers sensitive to hormones also involve the activation of G protein-coupled receptor and nuclear-mediated receptors that triggers multiple cellular signalling events to cause the disease. For example, the G protein-coupled estrogen receptor (GPER) plays an important role in cancer of both male and female reproductive systems[19]. Studies also highlighted the binding of nuclear receptors to their respective DNA target motifs across the genome, playing a critical role in the development and progression of cancer[20–22].

This growing evidence for the role of common gene and involvement of transmembrane and nuclear mediated signalling in tumorigenesis leads to the proposal of a combined analysis of multiple hormone-sensitive cancers (e.g., treating them as a single disease) to investigate the common genetic aetiology and identify new risk loci underlying the common pathway. However, human genomic studies of hormone-sensitive cancers have been limited to investigating site-specific cancers independently. While a few genome-wide association studies (GWASs) have provided evidence for a shared genetic basis between a limited number of cancer types (i.e., breast, prostate, endometrial and ovarian cancer)[23–25], it is unclear if the common germline genetic factors play a significant role in the shared mechanism of carcinogenesis[5,26].

Estimating SNP-based heritability can quantify the proportion of variance in disease liability explained by the genome-wide SNPs. When treating multiple hormone-sensitive cancers as a single disease, estimated SNP-based heritability can inform if the common germline variants contribute to the carcinogenic risk shared between hormone-sensitive cancers. Furthermore, analyses of shared heritability between the disease and other hormone-related phenotypes such as IGF-1, oestradiol and sex hormone binding globulin (SHBG) may provide information about the relationships between modifiable environmental risk factors and the risk of hormone-sensitive cancers.

The aim of this study is to estimate the SNP-based heritability for grouped hormone-sensitive cancers, using a broad definition including breast, prostate, uterine, ovarian, and thyroid cancers, among 15,197 hormone-sensitive cancer cases in a total of 288,837 participants in the UK Biobank (UKB). We also examine the genetic correlation between hormone-sensitive cancers and other non-cancer traits, with a view of establishing genome-wide level interactions. In this study, we show that a significant proportion of variance in hormone-sensitive cancers is explained by heritable genetic variants. Through the identification of genome-wide significant SNPs and analysis of genetic correlation, we uncovered molecular evidence of shared aetiology in hormone-sensitive cancers.

## Results

The characteristics of participants stratified by a group of cancer diagnoses are shown in Table 1. A total of 250,709 white Europeans were analysed in this study; including 15,197 (6.06%) hormone-sensitive cancer cases. In summary, 53.8% ($n = 155,392$) of the study samples were women, 93.48% ($n = 270,014$) were current alcohol drinkers, 42.5% ($n = 122,628$) were overweight, 54.59% ($n = 157,690$) had never smoked and 35.2% ($n = 101,521$) were previous smokers. There was a total of 21,973 incident cancer cases [diagnosed with cancer after baseline during follow-up] with a median follow-up year of 7.7 years (interquartile range [IQR] = 7.08–8.4) and 24,438 prevalent cancer cases (diagnosed with cancer before baseline assessment).

**SNP-based Heritability (SNP-$h^2$) for Groups of Cancers**. All grouped cancer (prevalent and incident) cases were included for the estimation of SNP-based heritability using individual-level data[27,28]. Here, we used a Genomic Restricted Maximum Likelihood (GREML) analysis, which is a statistical method that estimates the proportion of variance on one or more phenotypes attributed by all genetic polymorphisms using individual-level data to estimate the variance explained by all genetic polymorphisms (SNP-based heritability). We also used GWAS summary statistics to estimate SNP-based heritability, applying summary-level data[29]. In both approaches, the estimated heritability was transformed from the observed scale to the liability scale[28], assuming that the population lifetime prevalence of the group of cancers was the same as the proportion of cases in the sample used in this study. From the estimates in Fig. 1, it is apparent that the SNP-based heritability estimated in GREML for hormone-sensitive cancers was the highest ($h^2 = 10.06\%$ (se = 0.70%), $P = 2.11E-46$). In addition, the SNP-based heritability was examined for overall cancers and by grouping those cancer related to obesity. Significant heritability estimates for other cancer subgroupings was also observed, e.g., obesity-related cancer ($h^2 = 5.26\%$ (se = 0.47%), $P = 4.56E-28$); overall cancer ($h^2 = 4.38\%$ (se = 0.31%), $P = 3.27E-44$); non-hormone-sensitive cancer ($h^2 = 3.06\%$ (se = 0.72%), $P = 2.15E-05$); and non-obesity-related cancers ($h^2 = 1.69\%$ (se = 0.48%), $P = 4.66E-04$). The SNP-based heritability estimate using summary-level data shows a similar pattern of heritability estimates for all the sub-groupings of cancer (Supplementary Tables 1–5).

We also restricted the analysis to incident cancer cases only in the UKB. Similarly, heritability estimates in the liability scale for hormone-sensitive cancers were consistently higher than any other group of cancers when using incident cases only ($h^2 = 5.92\%$, se = 1.10%, $P = 7.84E-08$ for GREML and $h^2 = 5.60\%$, se = 1.58%, $P = 3.94E-04$ for LDSC) (Supplementary Table 5). The heritability estimates for non-obesity related cancers were not statistically significant in both methods ($h^2 = 0.43\%$, se = 0.75%, $P = 5.67E-01$ for GREML and $h^2 = 0.97\%$, se = 2.50%, $P = 6.98E-01$ for LDSC). In contrast, we observed a significant but lower heritability estimates for incident overall cancer cases ($h^2 = 3.1\%$, se = 0.48%, $P = 9.29E-11$ for GREML and $h^2 = 1.84\%$, se = 0.72%, $P = 1.06E-02$ for LDSC) (Supplementary Fig. 1).

**Table 1 Descriptive statistics for overall cancer, obesity-related and hormone-sensitive cancers in the UK Biobank ($N = 276,028$).**

| Characteristics | Controls*, N (%) 235,512 | Overall cancer cases, N (%) | Obesity related cancer cases‡, N (%) | Hormone-sensitive cancer cases‡ N (%) |
|---|---|---|---|---|
| Sex | | | | |
| Male | 110,150 (86.8) | 16,725 (13.2) | 12,504 (9.4) | 5730 (4.3) |
| Female | 125,362 (84.1) | 23,791 (15.9) | 13,602 (8.8) | 9467 (6.0) |
| Age at initial assessment | | | | |
| 39–49 years | 56,854 (24.1) | 4595 (11.3) | 1615 (2.4) | 666 (0.98) |
| 50–59 years | 81,500 (34.6) | 11,580 (28.6) | 7976 (7.9) | 4696 (4.7) |
| 60–73 years | 97,158 (41.2) | 24,341 (60.0) | 20,970 (15.6) | 12,481 (10.0) |
| BMI (kg/m²) | | | | |
| Underweight[<18.5 kg/m²] | 1189 (0.5) | 230 (0.5) | 132 (8.8) | 75 (5.0) |
| Normal[18.5–25 kg/m²] | 77,882 (33.0) | 13,0551 (32.2) | 7841 (8.2) | 4818 (5.0) |
| Overweight[25–30 kg/m²] | 99,872 (42.4) | 17,153 (42.3) | 11,408 (9.3) | 6575 (5.4) |
| Obese[≥30 kg/m²] | 55,833 (23.7) | 9929 (24.5) | 6638 (9.7) | 3688 (5.4) |
| Missing | 736 (0.3) | 149 (0.3) | 87 (9.0) | 41 (4.2) |
| Smoking status | | | | |
| Never | 130,787 (55.5) | 19,867 (49.0) | 12,944 (8.2) | 8113 (5.1) |
| Former | 80,716 (34.2) | 16,032 (39.6) | 10,624 (10.5) | 5833 (5.8) |
| Current | 23.255 (9.9) | 4432 (11.0) | 2416 (8.5) | 1189 (4.2) |
| Missing | 754 (0.3) | 185 (0.4) | 122 (11.8) | 62 (6.0) |
| Alcohol consumption | | | | |
| Never | 7070 (3.0) | 1355 (3.3) | 916 (10.4) | 590 (6.7) |
| Former | 7746 (3.3) | 1622 (4.0) | 1007 (10.4) | 512 (5.3) |
| Current | 220,529 (93.6) | 37,526 (92.6) | 24,171 (8.9) | 14,091 (5.2) |
| Missing | 167 (0.07) | 23 (0.06) | 12 (4.7) | 4 (1.5) |
| Educational status | | | | |
| None | 36,048 (15.3) | 8174 (20.2) | 5464 (11.7) | 3010 (6.5) |
| NVQ/CSE/A-levels | 107,162 (45.5) | 17,548 (43.3) | 10,974 (8.4) | 6339 (4.9) |
| Degree/professional | 90,481 (38.4) | 14,400 (35.5) | 9385 (8.5) | 5693 (5.2) |
| Missing | 1821 (0.7) | 394 (0.9) | 283 (11.9) | 155 (6.5) |

*BMI* Body Mass Index
*Controls are individuals without any cancer record in the cancer registry and who have had no self-report of cancer.
‡Obesity-related cancer includes postmenopausal breast cancer, prostate cancer, colon & rectal cancer, liver, stomach, pancreatic, oesophagus, thyroid, gallbladder, meningioma, ovary, uterus, kidney and multiple myeloma.
‡Hormone-sensitive cancers are those hormones sensitive cancers that include breast, prostate, uterine, ovarian, and thyroid cancers.

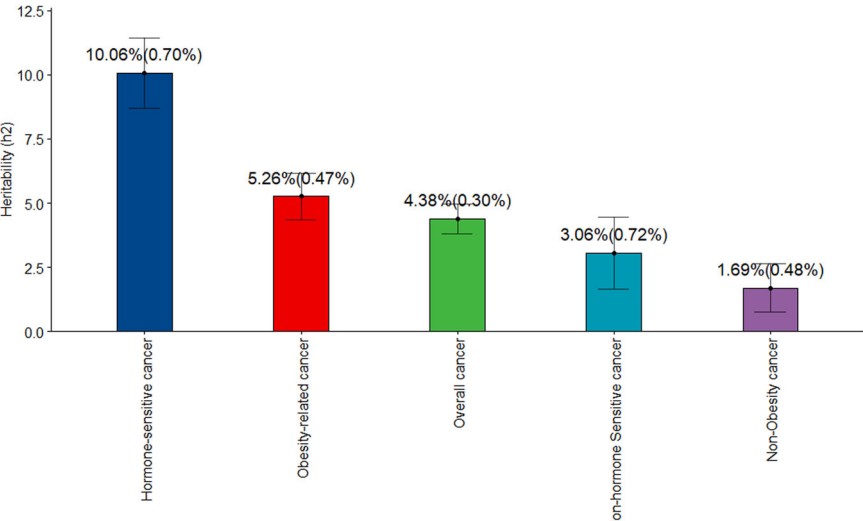

**Fig. 1 Estimated SNP-based heritability of grouped cancers using greml approach in the UKB.** It is shown that the $h^2$ for hormone-sensitive cancers is the highest among the other groups of cancers. The error bars are the 95% confidence interval of the estimates. The y axis shows the heritability estimates for each grouped cancer in x-axis.

**Genome-wide common SNPs association study (GWAS) for hormone-sensitive cancers**. The heritability estimates for hormone-sensitive cancers were consistently shown to be significant and higher than the other cancer subgroups across all methods applied in the liability scale for both scenarios (i.e., all cancer cases and incident cancer cases only). This clearly suggests that a significant proportion of phenotypic variation in hormone-sensitive cancer is explained by the aggregated effects of inherited

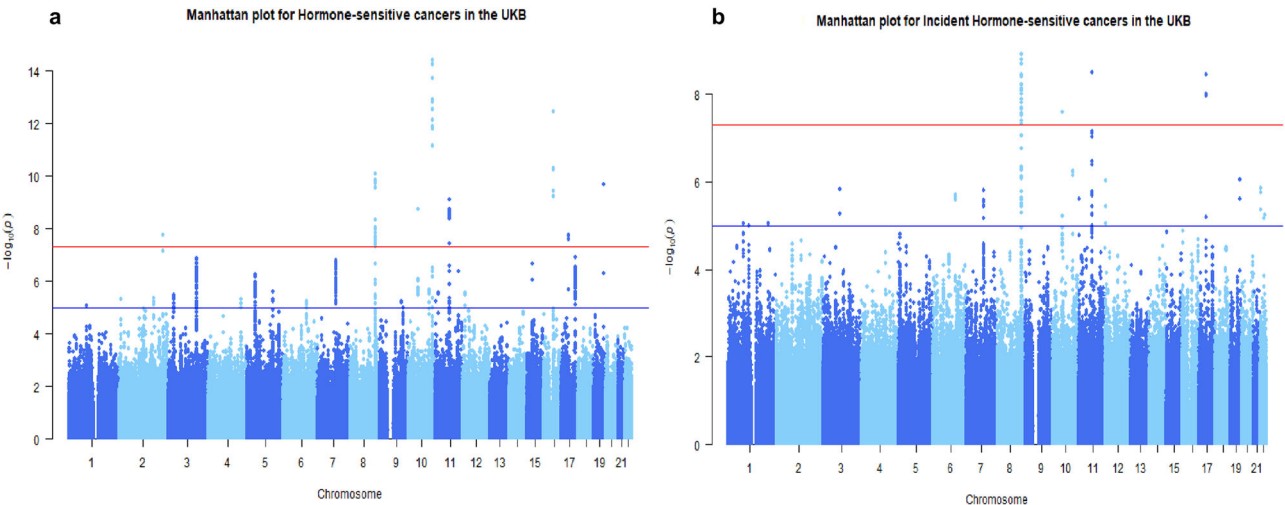

**Fig. 2 Manhattan plot for the GWAS analysis of the combined hormone-sensitive cancers in the UKB.** The plot shows on the Y-axis the negative log-base-10 of the *P* value for each of the SNPs positioned along the *X* axis in genomic order by chromosomal position. The red line shows the threshold for genome-wide significance ($P < 5 \times 10^{-8}$). SNPs with the lowest P value of significance (i.e., highest association with hormone-sensitive cancer) are positioned at the top of the graph. **a** The genome wide significant SNPs for all cases of hormonal cancers [incident and prevalent cases included]. **b** The panel is for incident cancer cases only. The list of genetic markers for each analysis is attached in the supplementary files (Supplementary Tables 7 and 8). The genomic inflation factor ($\lambda$) was rescaled for an equivalent study of 1000 cases/1000 controls ($\lambda_{1000}$ (all cases) = 1.003 and $\lambda_{1000}$ (incident cases) = 1.003).

genetic factors. We further carry out GWAS using genome-wide common SNPs, to identify genetic variants that are associated with hormone-sensitive cancer risk (see method).

We combined heterogeneous cancers that share a characteristics mechanism of carcinogenesis that involve hormones into a single phenotype of hormone-sensitive cancer, totalling 15,197 cases (combined prevalent and incident) and 235,512 controls. The number for each type of hormone-sensitive cancer is in Supplementary Table 6. Interestingly, our primary GWAS of grouped hormone-sensitive cancer uncovered 55 genome-wide significant variants that are associated with the risk of hormone-sensitive cancer at the genome-wide significant level of $p < 5 \times 10^{-8}$ (Fig. 2). This analysis demonstrated the existence of shared genetic variants across the different cancer types grouped as hormone-sensitive cancers. For these genetic variants, we replicated 36 independent SNPs associated with the risk of a specific type of cancer, such as breast, prostate, uterine or ovarian cancer, which were identified in previous GWAS[30–36]. Significant signals for each independent hormone-sensitive cancer involves 8q24.1, 10q26.13, 11q13.3, 16q12.1, and 17q12 genomic regions, of which 12 were for breast, 18 for prostate, 2 for uterine cancer, and 1 for ovarian cancer corresponded to previously identified as causal variants involved in the components of hormone-sensitive cancer (Supplementary Table 7). Overall, we found 12 independent association signals at chromosome 2, 8, 10, 11, 16, 17, and 19 out of the 55 genome-wide significant variants. A summary of these genome-wide significant independent loci for hormone-sensitive cancers association results for the 12 lead SNPs is provided in Table 2 and the LD heatmap in Supplementary Fig. 2a.

In a GWAS analysis restricted to 7038 incident hormone-sensitive cancer cases only (i.e., excluding prevalent cases), we found that significant associations were reduced from 55 to 33 significant SNPs. For these significant loci, 16 SNPs were located in already known susceptibility regions for hormone-sensitive cancers among the white European population, but they were independent of previously reported variants. The remaining 17 SNPs were in regions previously found to be associated with hormone-sensitive cancers among white Europeans. A list of

SNPs identified from GWAS in hormone-sensitive cancers can be found in Supplementary Table 8. For incident hormone-sensitive cancers, we found 8 independent loci (Supplementary Fig. 2b) and the list is provided in Supplementary Table 9. It was noted that genomic inflation factors were close to 1 for both GWAS analyses with all cases and incident hormone-sensitive cancer cases ($\lambda_{1000}$(all cases) = 1.003 and $\lambda_{1000}$(incident cases) = 1.003) (Supplementary Fig. 3).

We meta-analysed the single-traits GWAS for each hormone-sensitive cancer to gain a detailed understanding of potential genome-wide significant variant overlap. The analysis identified 37 variants associated with the risk of hormone-sensitive cancer at the prespecified significant level (Supplementary Table 10 and Supplementary Fig. 4), which was less than the number of SNPs found in the analysis with combined hormone-sensitive cancers as a single disease. Furthermore, upon repeating the meta-analysis of single-traits GWAS for incident cases of hormone-sensitive cancers, no genome-wide significant SNPs were found. This supports the conceptual premise for combining hormone-sensitive cancers as a single disease, which is likely to be more powerful to gain insight on molecular evidence of shared aetiology than the meta-analysis of single trait GWAS.

**Phenotypic correlation between hormone-sensitive cancer and non-cancer traits.** For better understanding of the genetic basis of hormone-sensitive cancer, we first quantify the phenotypic correlation with non-cancer traits known to be associated with cancer risk. The non-cancer traits were glycaemic traits [blood glucose level, HbA1c, type 2 diabetes (T2D)]; anthropometric traits [Waist-Hip Ratio (WHR), body mass index (BMI), WHR-adj-BMI, Waist Circumference (WC), Height-standing, body fat percentage,)]; metabolic traits and lipid profiles [cholesterol, tri-glyceride, high density lipoprotein (HDL), low density lipoprotein (LDL), apolipoprotein A and B]; menstrual factors [menopausal status]; behavioural-lifestyle factors [alcohol consumption, smoking, educational status and Townsend deprivation index (TDI)]; and cardiac traits [systolic blood pressure, diastolic blood pressure, C-reactive protein (CRP), cardiovascular disease status as binary trait and vitamin D].

**Table 2 Independent loci LD for genome wide significant SNPs of hormone-sensitive cancers GWAS in the UKB.**

| No | CHR | BP | SNP | A1 | *P* | BETA | STAT | NMISS |
|----|-----|-----|-----|-----|-----|------|------|-------|
| 1 | 2 | 217905832 | rs13387042 | G | 1.550E-08 | −0.003852 | −5.656 | 237,570 |
| 2 | 8 | 128011937 | rs10086908 | C | 1.671E-10 | −0.00476 | −6.389 | 236,744 |
| 3 | 8 | 128093297 | rs1016343 | T | 2.511E-08 | 0.004711 | 5.573 | 237,567 |
| 4 | 8 | 128107153 | rs16901949 | C | 1.875E-08 | 0.01064 | 5.623 | 237,553 |
| 5 | 8 | 128407443 | rs10505477 | G | 3.035E-08 | −0.003777 | −5.540 | 237,573 |
| 6 | 8 | 128517573 | rs4242382 | A | 1.292E-08 | 0.006406 | 5.687 | 237,573 |
| 7 | 10 | 51549496 | rs10993994 | T | 2.434E-09 | 0.004161 | 5.966 | 237,568 |
| 8 | 10 | 123334457 | rs10736303 | G | 7.971E-12 | 0.004688 | 6.839 | 236,037 |
| 9 | 11 | 68973970 | rs4255548 | A | 3.927E-09 | −0.004171 | −5.887 | 231,722 |
| 10 | 16 | 52538040 | rs17271951 | C | 4.424E-11 | 0.005219 | 6.589 | 234,844 |
| 11 | 17 | 36098040 | rs4430796 | G | 2.362E-08 | −0.003817 | −5.583 | 237,530 |
| 12 | 19 | 51361757 | rs17632542 | C | 2.668E-10 | −0.008252 | −6.317 | 237,572 |

We observed a modest phenotypic correlation in selected glycaemic, cardiovascular and anthropometric traits. For example, there was a positive phenotypic correlation ($r_p$) between standing height and hormone-sensitive cancer ($r_p = 0.0130$, se $= 0.0020$, $P = 1.78E-10$) and WC ($r_p = 0.0106$, se $= 0.0020$, $P = 2.36E-07$). Further to this, we observed a significant positive correlation for disease traits that involves T2D ($r_p = 0.0084$, se $= 0.0020$, $P = 4.49E-05$) and negative correlation with cardiovascular disease ($r_p = -0.0090$, se $= 0.0020$, $P = 9.64E-06$). For the cancer related traits, a negative correlation was observed between hormone-sensitive cancers and oestradiol level ($r_p = -0.0190$, se $= 0.0022$, $P = 2.20E-16$); SHBG ($r_p = -0.0059$, se $= 0.0022$, $P = 7.35E-03$).

The most striking result to emerge from the phenotypic correlation data is that although there were similar patterns of significant correlations with most of the non-cancer traits in the analysis restricted to incident cases, some estimates were substantially changed. For example, we observed a substantially reduced positive phenotypic correlation between incident hormone-sensitive cancers and oestradiol level ($r_p = 0.0025$, se $= 0.0022$, $P = 2.68E-01$). Interestingly, a significant negative phenotypic correlation observed between incident hormone-sensitive cancers and APOA1 ($r_p = -0.0065$, se $= 0.0022$, $P = 3.83E-03$). The results of these analyses are summarised in Table 3.

**Genetic correlation between group hormone-sensitive cancers and non-cancer traits.** In further analysis to explain the shared genetic architecture of grouped hormone-sensitive cancers, we estimated the genetic correlation with the six non-cancer sub-group traits using GWAS summary statistics (Supplementary Table 11) for bivariate LDSC which is a fast and robust method[37] as a quick scan in the dataset and for those nominally significant traits using individual-level measurement in bivariate GREML. We estimate the genetic correlation between grouped hormone-sensitive cancers and some non-cancer traits using individual-level genotype data analysed in bivariate GREML. Interestingly, significant positive genetic correlations were observed between IGF-1 ($r_g = 8.43\%$, se $= 1.38\%$, $P = 1.10E-09$); standing height ($r_g = 4.32\%$, se $= 1.31\%$, $P = 9.59E-04$) and hormone-sensitive cancer that provides a suggestive clue to cancer aetiology wherein an increase in IGF-1 level and height confers a higher risk of hormone-sensitive cancer. Moreover, a marginally significant inverse genetic correlations were observed between hormone-sensitive cancers and three other non-cancer traits, namely serum oestradiol ($r_g = -40.86\%$, se $= 8.60\%$, $P = 2.02E-06$); calculated free oestradiol ($r_g = -6.68\%$, se $= 1.60\%$, $P = 3.15E-05$); SHBG ($r_g = -3.33\%$, se $= 1.92\%$, $P = 8.20E-02$)

and diastolic blood pressure (DBP) ($r_g = -4.40\%$, se $= 0.02116$, $P = 3.74E-02$) (Fig. 3).

In an analysis restricted to incident cancer cases, we observed a non-significant but positive genetic correlation for serum oestradiol ($r_g = 17.08\%$, se $= 14.56\%$, $P = 2.41E-01$) (Fig. 4) contrary to the negative genetic correlation estimate obtained when all combined cases were analysed together (Fig. 3). This suggests that the genetic effects of oestradiol may be positively correlated with the genetic risk of incidence of hormone-sensitive cancer[38], however, after the onset of hormone-sensitive cancer, the genetic association may be driven by a totally different mechanism, resulting in a negative genetic correlation. For standing height ($r_g = 9.01\%$, se $= 1.97\%$, $P = 4.93E-06$) and IGF-1 ($r_g = 12.13\%$, se $= 2.50\%$, $P = 1.31E-06$), the direction of estimated genetic correlation is consistent and always positive whether using all cases (Fig. 3) or incident cases only (Fig. 4). Apolipoprotein A ($r_g = 11.16\%$, se $= 2.58\%$, $P = 1.55E-05$) appeared to have a significant negative genetic correlation when using incident cases only, which was different from the result obtained with all cases, implying tumour suppressive role of Apolipoprotein A in the incidence of hormone-sensitive cancer development. Compared to all cases, we further noted a slightly significant and higher estimate of negative genetic correlation in calculated free oestradiol (rg $= -8.86\%$, se $= 2.80\%$, $P = 1.57E-03$); SHBG ($r_g = -8.78\%$, se $= 2.73\%$, $P = 1.32E-03$) and educational status ($r_g = -11.95\%$, se $= 4.85\%$, $P = 1.39E-02$) for incident cases. For diastolic blood pressure ($r_g = -2.06\%$, se $= 2.82\%$, $P = 4.62E-01$) a similar non-significant negative genetic correlation was observed even though the analyses for non-cancer traits were restricted to individuals who did not have cancer at baseline (Fig. 4).

In the analyses of genetic correlation using summary statistics in the UKB, though not statistically significant the estimates are mostly agreed with the individual level data estimates. The estimates for genetic correlation using summary statistics in bivariate LDSC are summarised and presented in Supplementary Table 11.

**Genetic correlation between cancers.** We further quantified the genetic correlation among the specific types of cancers in the group of hormone-sensitive cancers to see their shared genetic architecture. We used bivariate LDSC that is computationally efficient and not biased by sample overlap in two sets of case-control data between which controls are common[37]. In the pairwise comparison, we observed a positive genetic correlation between colorectal cancer and cancer of the kidney ($r_g = 0.3712$, se $= 0.2965$); women breast cancer and uterine cancer ($r_g = 0.3211$, se $= 0.1990$) although they were not significantly

**Table 3 Phenotypic correlation between hormone-sensitive cancers and other non-cancer traits in the UKB.**

| Traits | All hormone-sensitive cancer cases combined (incident and prevalent) | | | Incident hormonal cancers cases | | |
|---|---|---|---|---|---|---|
| | Phenotypic correlation | | | Phenotypic correlation | | |
| | $r_p$ | SE | p-value | $r_p$ | SE | p-value |
| Glycaemic traits | | | | | | |
| T2D | 0.0084 | 0.0020 | 4.49E-05* | 0.0072 | 0.0021 | 9.33E-04* |
| Glucose | 0.0022 | 0.0022 | 3.13E-01 | −0.0050 | 0.0022 | 2.85E-02* |
| HbA1c | −0.0015 | 0.0021 | 4.50E-01 | −0.0075 | 0.0021 | 4.89E-04* |
| Anthropometric traits | | | | | | |
| BMI | 0.0059 | 0.0020 | 4.05E-03* | 0.0054 | 0.0020 | 8.91E-03* |
| WHR | 0.0064 | 0.0020 | 1.78E-03* | 0.0033 | 0.0021 | 1.14E-01 |
| WHRadjBMI | 0.0053 | 0.0020 | 9.68E-03* | 0.0028 | 0.0021 | 1.70E-01 |
| WC | 0.0106 | 0.0020 | 2.36E-07* | 0.0068 | 0.0020 | 1.01E-03* |
| Height (standing) | 0.0130 | 0.0020 | 1.78E-10* | 0.0089 | 0.0021 | 2.11E-05* |
| Body fat percentage | 0.0058 | 0.0021 | 5.04E-03* | 0.0045 | 0.0021 | 3.41E-02* |
| Lipid profile | | | | | | |
| Cholesterol | 0.0045 | 0.0021 | 3.28E-02* | −0.0037 | 0.0021 | 8.26E-02 |
| Triglyceride | 0.0067 | 0.0021 | 1.45E-03* | −0.0059 | 0.0021 | 5.23E-03* |
| HDL | −0.0051 | 0.0022 | 2.10E-02* | −0.0054 | 0.0022 | 2.68E-02* |
| LDL | 0.0051 | 0.0021 | 1.39E-02* | −0.0014 | 0.0021 | 5.30E-01 |
| APOA1 | −0.0005 | 0.0022 | 8.21E-01 | −0.0065 | 0.0022 | 3.83E-03* |
| APOB | 0.0050 | 0.0021 | 1.61E-02* | −0.0017 | 0.0021 | 4.08E-01 |
| Behavioural-lifestyle | | | | | | |
| Alcohol | 0.0004 | 0.0080 | 8.22E-01 | −2.48E-11 | 0.0021 | 1.00E+00 |
| Smoking | −0.0001 | 0.0020 | 9.32E-01 | 2.24E-11 | 0.0021 | 1.00E+00 |
| Education | 0.0013 | 0.0020 | 5.16E-01 | 6.80E-11 | 0.0021 | 1.00E+00 |
| Townsend | −1.28E-05 | 0.0020 | 9.95E-01 | −1.00E-11 | 0.0020 | 1.00E+00 |
| Cardiac traits | | | | | | |
| Systolic Blood Pressure | 0.0018 | 0.0021 | 4.00E-01 | −0.0015 | 0.0021 | 4.93E-01 |
| Diastolic Blood Pressure | 0.0067 | 0.0021 | 1.48E-03* | 0.0023 | 0.0021 | 2.74E-01 |
| Cardiovascular Disease | −0.0090 | 0.0020 | 9.64E-06* | −0.0061 | 0.0021 | 3.35E-03* |
| C-reactive Protein | 0.0080 | 0.0021 | 1.46E-04* | −0.0005 | 0.0021 | 8.16E-01 |
| Vitamin D | 0.0048 | 0.0021 | 2.40E-02* | 0.0048 | 0.0021 | 2.68E-02* |
| Menstrual factors | | | | | | |
| Menopausal Status | 0.0009 | 0.0035 | 7.97E-01 | 0.0077 | 0.0036 | 3.48E-02* |
| Cancer-related | | | | | | |
| SHBG | −0.0059 | 0.0022 | 7.35E-03* | −0.0086 | 0.0022 | 1.20E-04* |
| Testosterone | −0.0050 | 0.0022 | 2.13E-02* | 0.0068 | 0.0022 | 2.15E-03* |
| Oestradiol | −0.0190 | 0.0022 | 2.20E-16* | 0.0025 | 0.0022 | 2.68E-01 |
| IGF-1 | 0.0094 | 0.0021 | 7.30E-06* | 0.0102 | 0.0021 | 2.00E-06* |

An asterisk indicates significance with $P < 0.05$ using two tailed hypothesis test and normal distribution of the Fischer transformed correlation coefficient. The estimates are reported with their respective standard error.
$r_p$ phenotypic correlation, $r_g$ genotypic correlation, SE standard error, T2D type II diabetes, HbA1c glycate haemoglobin, BMI body mass index, WHR waist to hip ratio, WC waist circumference, HDL high density lipoprotein, LDL low density lipoprotein, ApoA1 apolipoprotein A 1, ApoB apolipoprotein B, SHBG Sex hormone binding globulin.

different from zero. We also found a negative, but non-significant, genetic correlation between prostate cancer and colorectal cancers ($r_g = −0.1073$, se = 0.1314); uterine cancer and multiple myeloma ($r_g = −0.1474$, se = 0.5053) (Fig. 5). Although none of the estimated genetic correlations were significantly different from zero i.e., showing there is not a significant linear correlation to one another, most estimates were significantly different from 1 or −1, indicating that these types of cancers are genetically heterogeneous.

**Leave-one-out (LOO) analysis approach for hormone-sensitive cancers.** We conducted an iterative leave-one-out (LOO) analysis that involves a different combination of hormone-sensitive cancers (Supplementary Fig. 5). There was a significant modest genetic correlation in the leave-one-out analysis between each component of the hormone-sensitive cancers. For example, we observed a modest positive genotypic correlation between female breast cancer and grouped hormone-sensitive cancer without female breast cancer ($r_g = 0.1662$, se = 0.0930); prostate cancer and grouped hormone-sensitive cancer excluding prostate cancer ($r_g = 0.2209$, se = 0.1101); uterine cancer ($r_g = 0.3487$, se = 0.1889) and grouped hormone-sensitive cancers without uterine cancer. For ovarian and thyroid cancer, since the number of cases was not sufficient for bivariate LDSC regression analysis, we excluded the two hormone-sensitive cancers from the leave-one-out analysis (Table 4).

We further carried out genetic correlation analyses into grouped hormone-sensitive cancers and other obesity-related non-hormone-sensitive cancers in the UKB (namely colorectal, renal, and multiple myeloma) to gain more detailed understanding of the complexities of hormone-cancer phenomena. Hormone-sensitive cancers appeared to have a negative genetic correlation with cancer of kidney ($r_g = −0.0786$, se = 0.2362); positive genetic correlation with colorectal cancer ($r_g = 0.1551$, se = 0.1254) and multiple myeloma ($r_g = 0.1129$, se = 0.2056) (Table 4).

The genetic correlation between multiple myeloma and hormone-sensitive cancers excluding breast cancers demonstrated a positive genetic correlation ($r_g = 0.1926$, se = 0.2295).

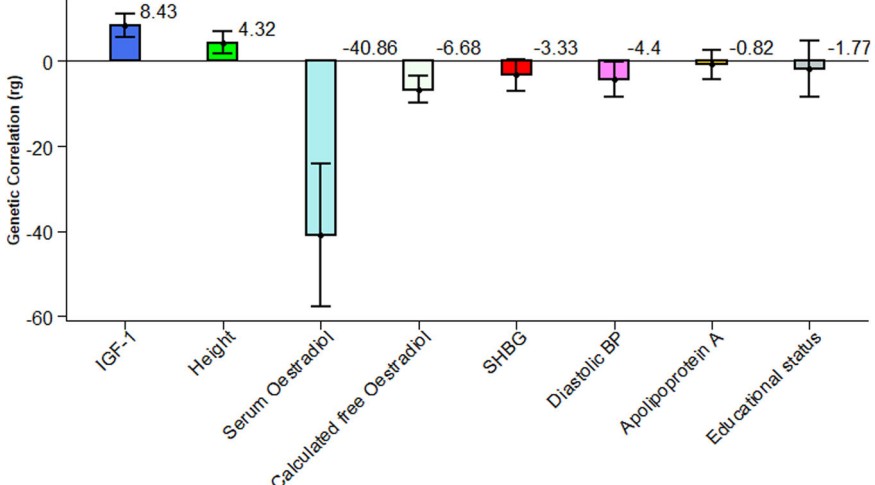

**Fig. 3 Genetic correlation between all hormone-sensitive cancers [Prevalent and Incident] and non-cancer traits using bivariate GREML in the UK Biobank.** The values are in percentage. SHBG Sex Hormone Binding Globulin, IGF-1 Insulin Like growth factor. The error bars are indicating the 95% CI of the estimates.

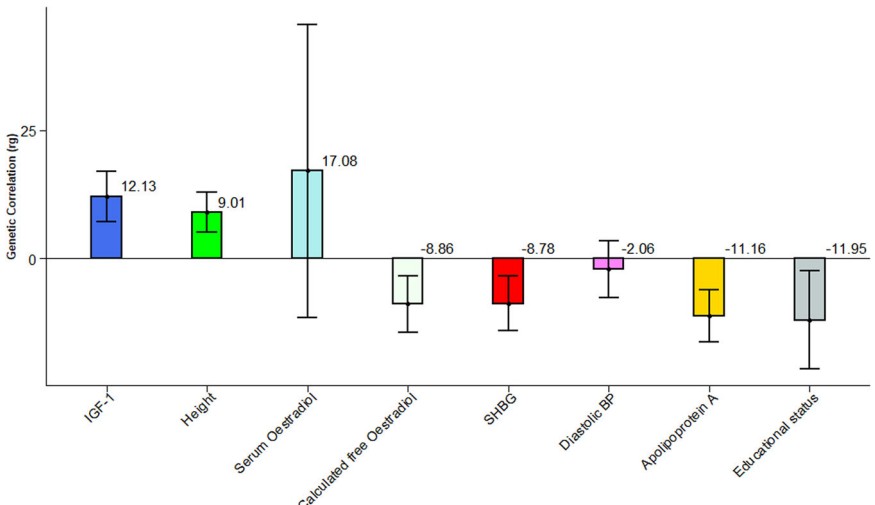

**Fig. 4 Genetic correlation between incident hormone-sensitive cancers and non-cancer traits using bivariate GREML in the UK Biobank.** The values are in percentage. SHBG Sex Hormone Binding Globulin, IGF-1 Insulin Like growth factor. The error bars are indicating the 95% CI of the estimates.

We observed a higher genetic correlation between hormone-sensitive cancer without prostate cancer and colorectal cancer ($r_g = 0.3061$, se = 0.1597). Hormone-sensitive cancers without uterine cancer demonstrated a higher genetic correlation with colorectal cancer ($r_g = 0.1666$, se = 0.1229). None of the genetic correlations estimated in these analyses were statistically significant probably due to lack of power. Taken together, while these estimated genetic correlations suggest a common pathway in the aetiology of hormone-sensitive cancer, there is significant evidence of genetic heterogeneity among the cancer types (Table 4).

While these estimates suggest that there is significant genetic heterogeneity among these cancers, the estimated SNP-based heritability of the overall hormone-sensitive cancer coded as a single disease shows that the phenotypic variance explained by the common genetic factors is significantly different from zero (Fig. 1).

**Gene-environment interaction (GxE) for selected environmental traits.** Finally, we investigated the gene-environment interaction, using the hormone-sensitive cancers as the main phenotypes and metabolic health-related traits as environmental variables. Note that we used incident cases only for this gene-environment interaction analysis. The hormone-sensitive cancer phenotype status was adjusted for multiple variables that include assessment centre, batch effect, birthplace, age, sex, educational status, the first 10 principal components, smoking status, alcohol consumption, and TDI. Given the characteristics of these environmental variables, we have applied the bivariate GREML or GxEsum method[39]. The baseline BMI measurement is categorised as normal and higher based on the World Health Organisation (WHO) BMI threshold recommendations[40]; metabolic markers classified as favourable and unfavourable metabolic environment from the metabolic subgroup analysis in the UKB using machine-learning data-driven analysis[41] and sex as a discrete variable were analysed in bivariate GREML. This bivariate GREML analysis was applied to detect the interaction using individual-level measurement in the UKB.

In the bivariate GREML analysis that requires individual-level genotype data, sex, BMI, and metabolic environment were included as an environment to detect their role in the aetiology

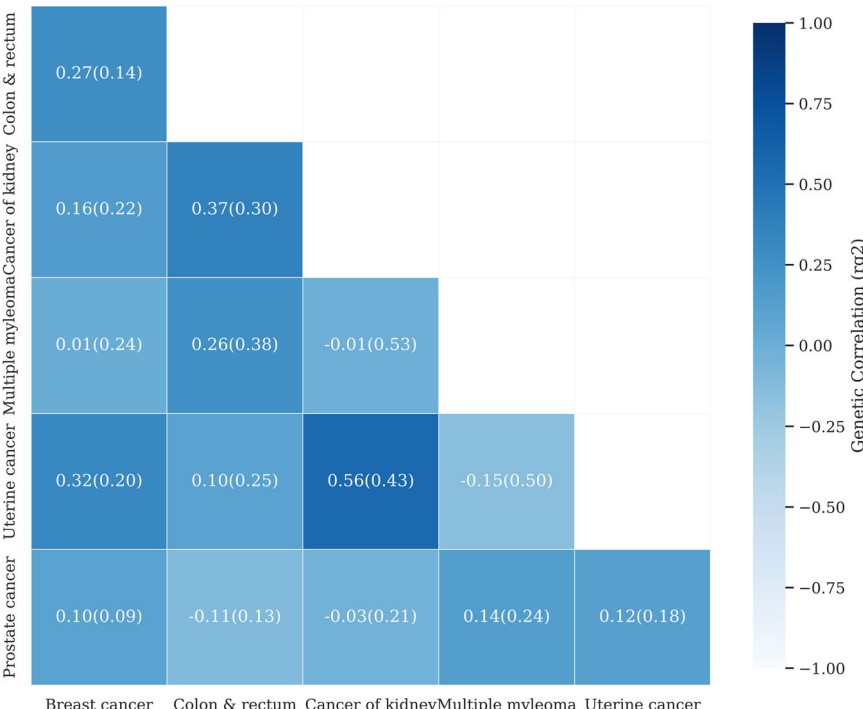

**Fig. 5 Estimation of pair-wise genetic correlation among obesity-related cancers in which breast, prostate and uterine cancers are hormone-sensitive cancers.** The positive genetic correlations are colorectal cancer with cancer of the kidney, women breast cancer with uterine cancer. The negative genetic correlation includes prostate cancer with colorectal cancer, uterine cancer with multiple myeloma. The estimates with the standard error ($r_g \pm se$) are obtained applying the bivariate LDSC method. Ovarian and thyroid cancers were not estimable, which was probably due to the fact that the number of cases was not sufficient for LDSC in the analysis of these diseases.

of hormone-sensitive cancers. In using the GREML method for BMI classified as normal and higher, significant evidence for GxE interaction was found as the genetic risk of hormone-sensitive cancer was heterogeneous between the two environments. Estimated genetic correlation was significantly different from 1 ($P = 6.00E\text{-}05$ in Table 5). Likewise, the estimated genetic correlation between favourable and unfavourable metabolic environments was also significantly different from 1 ($P\text{-value} = 1.87E\text{-}03$), indicating a significant GxE interaction. Although there is significant heterogeneity between males and females in the genetic risk of hormone-sensitive cancers when sex is included as environment, the observed genetic heterogeneity may not be because of the gene by sex (GxSex) interaction, given the diversified nature of distinct cancer types, each of which included is predominantly female or male-only cancer. Therefore, the finding reflects the genetic heterogeneity between sex-specific cancers (as shown in Table 5) as a result of diversified cancers, and it is not conclusive that the genetic risk of hormone-sensitive cancers is modulated by sex as an environment.

Further for quantitative environmental traits that include central obesity measured by WC, TDI, Apolipoprotein B, IGF-1, and physical activity measured in minutes per week were analysed using the GxEsum method based on GWAS summary statistics[39]. In the analyses, we did not find any significant GxE variance (Supplementary Table 12).

## Discussion

A growing number of population-based genomic studies have emphasised the role of hormones and their metabolites in modifying gene-phenotype pathways of cancers[8]. In the current study, we conducted a comprehensive analysis to estimate SNP-based heritability, a GWAS that focused on grouped hormone-sensitive

cancer and estimated the phenotypic and genetic correlation with other non-cancer traits in a large contemporary cohort. This study confirms that genome-wide common SNPs contribute to a substantial proportion of the phenotypic variance of hormone-sensitive cancers. In contrast, a relatively small proportion of phenotypic variance is captured by genome-wide common SNPs for non-hormonal cancers. A cross-cancer GWAS approach was applied to hormone-sensitive cancers in which we identified multiple genome-wide significant SNPs that had common effects shared between hormone-sensitive cancers. Interestingly, there was also significant genetic heterogeneity among hormone-sensitivity cancers, i.e., estimated genetic correlation for a pair of hormone-sensitivity cancers was significantly different from 1. We also found that the hormone-sensitive cancer status was significantly associated with non-cancer traits, e.g., IGF-1 and height signifying the suggestive role of these non-cancer traits in the complex biology of cancer.

In the current study, we applied GREML and LDSC methods of estimating heritability in which the GREML estimates were higher than LDSC. The variation demonstrated wherein the GREML analysis in the liability scale showed a 10% of phenotypic variability in hormone-sensitive cancer is due to genetics, further suggesting the existence of shared underlying biology for the combined hormone-sensitive cancers. This further suggests that previous site-specific independent cancer heritability estimates explain a small fraction of the shared heritability, and a fraction of this heritability can be explained by genome-wide common SNPs without the need for other variants such as structural and rare variants in whole-exome and whole-genome sequencing. In contrast to earlier findings, however, our heritability estimate is substantially lower than summary statistics-based estimates for each component of site-specific hormone-sensitive cancer that ranges from 7% (ovarian) to 27% (prostate) on a liability scale[42].

**Table 4 Genetic correlation using a leave-one-out analysis approach for hormone-sensitive cancers using bivariate LDSC in the UKB.**

| Cancer types | Women breast cancer | Prostate cancer | Uterine cancer | Colon & rectum | Cancer of kidney | Multiple myeloma |
|---|---|---|---|---|---|---|
| Hormonal cancer (as a single disease) | 0.6796 (0.0568) | 0.7759 (0.0474) | 0.4522 (0.1637) | 0.1551 (0.1254) | −0.0786 (0.2362) | 0.1129 (0.2056) |
| Hormonal cancer excluding breast cancer | 0.1662 (0.0930) | | | −0.0152 (0.1241) | −0.0088 (0.2290) | 0.1926 (0.2295) |
| Hormonal cancer excluding prostate cancer | | 0.2209 (0.1101) | | 0.3061 (0.1597) | 0.0645 (0.2853) | 0.0526 (0.2508) |
| Hormonal cancer excluding ovarian cancer | | | | 0.1208 (0.1247) | −0.1960 (0.2324) | 0.0692 (0.2031) |
| Hormonal cancer excluding uterine cancer | | | 0.3487 (0.1889) | 0.1666 (0.1229) | −0.1117 (0.2360) | 0.1520 (0.2096) |
| Hormonal cancer excluding thyroid cancer | | | | 0.1629 (0.1289) | −0.0561 (0.2386) | 0.0605 (0.2108) |

Hormone-sensitive cancer type includes five cancers namely women breast cancer, prostate, ovarian, uterine, and thyroid cancer. The genetic correlation involving ovarian and thyroid cancers were not estimable in the LDSC, probably because the numbers of cases for the two cancers were not sufficient (results not shown).

**Table 5 GREML based GxE interaction estimates for incident hormone-sensitive cancers using baseline measured traits.**

| Environment | $h^2$(se) | $r_g$(se) | P-Value |
|---|---|---|---|
| Body Mass Index (BMI) | | | |
| BMI Normal | 8.31% (2.95%) | 11.91% (21.95%) | 6.00E-05* |
| BMI High | 7.69% (1.37%) | | |
| Metabolic environment | | | |
| Favourable | 6.20% (1.61%) | 29.78% (22.5%) | 1.87E-03* |
| Unfavourable | 14.40% (2.75%) | | |
| ‡Sex | | | |
| Female | 9.10% (0.89%) | 14.82% (5.71%) | 9.42E-03* |
| Male | 22.37% (1.34%) | | |

$h^2$ heritability, $r_g$ genetic correlation, se standard error. The baseline BMI measurement is categorised as normal 18.5–25 kg/m$^2$ and higher BMI (≥25 kg/m$^2$); baseline biomarkers and anthropometric measurement as favourable and unfavourable to metabolic health consequences. For BMI and metabolic health environment all incident cases are used.
‡For sex as heterogeneous environment all hormone-sensitive cancer (incident and prevalent) cases are included.

There are two likely causes for the discrepancy between heritability estimates in the current study and previous studies. First, the difference could be attributed to the genetic heterogeneity of the combined cancers as evidenced in our genetic correlation estimates between cancers. Therefore, a reduced heritability is expected when these genetically heterogeneous cancers are grouped as a single trait. Second, the discrepancy can be explained in part by the difference in the level of information used wherein individual-level data from the UKB is used in our estimate whereas previous studies used GWAS summary statistics with a greater number of cases owing to higher heritability estimates. Although the estimates are low as compared to previous site-specific cancer components, our finding, however, provides a comprehensive analysis suggesting a through reconsideration of cancer classification for shared biological mechanism of carcinogenesis.

The analytical performance of GWAS is highly dependent upon the size of the cohort and the degree of phenotypic similarity of the combined traits[43]. Therefore, cross-trait GWAS recently adapted to identify common factors of interest in precision medicine that involves identification of genetic susceptibility loci for inflammatory bowel disease, mostly shared between Crohn's disease and Ulcerative colitis[44], and among five major psychiatric disorders generating quantified molecular evidence for the need to investigate common pathophysiology for related disorders[45,46]. Despite overwhelming success in other medical fields, cross-traits analysis has not been widely applied in cancer genetics. Furthermore, based on the GWAS to date on cancer, many independent cancer susceptibility variants have been identified. When these variants are combined into polygenic risk scores, they explain a small fraction of the heritability of cancer and show differential associations by tumour subtypes. However, it is only a few studies have combined some site-specific hormone-sensitive cancers[23,25]. Therefore, when cross-trait effects exist, the current study has important implications to systematically integrate the phenome-wide data available for genetic association analysis with improved statistical power in detecting significant genetic loci for meaningful biological interpretation. Moreover, our hypothesis of common shared aetiology is supported by the finding that multiple SNPs in the FGFR-2 and POU5F1B gene region are associated with hormone-sensitive cancers implying the presence of commonly expressed genomic regions. Evidence suggests that gene overexpression may lead to increased angiogenesis and autocrine stimulation of cancer cells[47]. For example, there is strong evidence that FGF-2 ligand and FGFR-2 receptors are important in breast cancer

tumorigenesis[10,48]. POU5F1B has been suggested to be involved in prostate cancer pathogenesis[17]. It has been demonstrated that hormones can express FGF-2 and POU5F1B genes and the level is progressively elevated during initial phases of tumorigenesis, and further its expression is higher in cancerous tissue in comparison with adjacent normal tissue or benign ones[12,14].

Most cancer genomics research is focused on somatic events, such as acquired mutations; but increasing evidence suggests that germline variants have been experimentally demonstrated to play a significant role in cancer risk prediction[49] and may also inform decisions about cancer-directed therapy[50]. Therefore, in the current study detecting common genetic variants across major cancers that shared similar aetiologic pathways will facilitate our understanding of the possible shared genetic basis of these cancers to develop more optimised diagnostic criteria. Our multi-trait GWAS analysis can be used to look for germline variants and understand how specific genetic variants may contribute to a broad spectrum of illness and provide information about the degree to which these disorders may have a shared genetic risk factor. To the extent that these genes may have broad effects, they could be potential targets for developing new treatments that might help treat multiple cancer conditions. In agreement with our findings, previous studies have implicated these genes in liability to each site-specific cancer in different population[51–54]. This supports the implementation of such combined analysis that provides more insight in the complex pathway underlying hormone-sensitive cancer biology with the expected molecular evidence on shared genetic risk factors seen in previous studies of major psychiatric and inflammatory disorders. This molecular evidence of shared genetic influence in hormone-sensitive cancers can be extended to design public health intervention addressing multiple cancers at affordable cost in genetic screening.

Epidemiological studies have identified an association between height, IGF-1, oestradiol, and cancer incidence to provide clues to cancer aetiology. The risk of IGF-1 in cancer is further established in deciphering the mechanism of height to cause cancer[55,56]. In the current study, we found a positive genetic correlation in all cases of hormone-sensitive cancers with IGF-1 and standing height, which suggests there is an increased correlation in height and IGF-1 concentration to develop hormone-sensitive cancer. The observed correlation between standing height and caner development might be explained by the increased standing height that reflects more stem cells as a risk of acquiring mutations during cell division over time, and further circulating level of IGF-1 as the major determinants of height[57]. There can be, however, other possible explanations. In contrast, serum oestradiol level showed a negative genetic correlation with all cases of hormone-sensitive cancers suggesting the presence of lowered risk. However, in the analysis restricted to incident hormone-sensitive cancers cases, serum oestradiol exhibited a positive genetic correlation (although not significant), suggesting that exposure to ovarian steroids increases the risk of developing hormone-sensitive cancers[58]. The current study further revealed that the positive genetic correlation of height and IGF-1 with cancer remained positively significant with incident cases suggesting that the correlation is related to the commonality within the combined cancers in their gene alteration and gene expression pattern.

Contrary to expectations for the rest of the traits, this study did not find a statistically significant genetic correlation between non-cancer trait subgroups and hormone-sensitive cancers. From previous epidemiological studies, it has been suggested that there is a correlation between non-cancer traits and specific hormone-sensitive cancers. This does not appear to be the case in our analysis. The observed low correlation can be explained in part by the underpowered nature of the current study to detect a phenotypic and genetic correlation between hormone-sensitive cancers and non-cancer traits. Therefore, non-genetic factors could be a major reason, if not the only one, significantly explaining the phenotype variance in cancer. Although the estimated genetic correlations are low, they can still be used as a training set in genomic risk prediction to improve the accuracy. In genomic risk predictions when traits were combined as a single trait, slightly increased prediction accuracy was observed[59,60]. This suggests that substantial improvements in predictive power are attainable using training sets of combined cancer with molecular evidence of shared genetic contribution.

Apart from considering the correlation of variables, detecting the interaction with the environment may have an important implication in clinical care[61,62]. Globally, the incidence of cancer has been steadily increasing for the past decades mirroring an increase in the prevalence of obesity[1]. The genetic effects of hormone-sensitive cancers can be modulated by obesity. Therefore, we sought to estimate the gene-environment interaction to shed light on the causal relationships of modifiable environmental risk factor such as BMI and hormone-sensitive cancers. Further, we found significant interaction between genetics and adiposity-related factors as environment to interact with and modulate the development of hormone-sensitive cancers. The nominally significant GxSex interaction observed cannot be fully attributed to the gene-interaction effect of sex since this might have occurred as a result of unequal distribution of hormone-sensitive cases by sex, i.e., majority of the grouped cancers are female dominated cancer types. Although the combined cancers demonstrated a shared aetiology, the pairwise genetic correlation comparison evidenced that they are heterogeneous. i.e., the five hormone-sensitive cancers have their unique pathogenic variants besides the shared genes. There is also further heterogeneity within the site-specific cancers. Endometrial cancer, for example, is a heterogeneous cancer that is believed to have two biologically different subtypes that exhibit a different mechanism of tumorigenesis and disease progression[63].

A major strength of the present study is that it constitutes a greater number of hormone-sensitive cancers grouped to better understand the complex underlying pathway of the disease biology. Previous studies were focusing on each site-specific hormone-sensitive cancer independently. Further, information on non-cancer traits was used from the large dataset of the UKB. This study offers significant insights into the heritability estimates of hormone-sensitive cancer. However, our findings should be interpreted in light of the limitations. First, participants in the UK Biobank are restricted to middle and old age, which is not representative of the general population on a variety of socio-demographic, lifestyle, and health-related characteristics, with evidence of a "healthy volunteer" selection bias[64]. Second, while the total sample size was large for the grouped cancer, the number of cases for some specific hormone-sensitive cancers (e.g., uterine and thyroid) could be limited resulting in a large standard error for genetic correlation analysis. Therefore, further studies with a larger sample size for each cancer are warranted to validate our results. Third, in our report of heritability in a liability scale, we assumed the population level prevalence of the disease trait is identical to the observed sample prevalence, but the disease prevalence such as cancer in the UKB is often lower than population prevalence as the dataset is not representative of the UK population[64]. Further, we did not classify subtypes of breast cancer, such as estrogen receptor [ER] positive/negative, progesterone receptor [PR] positive/negative, and triple-negative breast cancers, in our analyses because no such information was available to us. Finally, the present study was conducted in a population of European genetic ancestry, so the generalisability of our findings to other ancestry populations is limited.

In conclusion, we show that common genetic factors are a part to play in the mechanism of carcinogenesis shared by hormone-sensitive cancers, evidenced by the fact that SNP-based heritability is substantial and there are 55 genome-wide significant variants when combining multiple hormone-sensitive cancers as a single disease. Albeit these common genetic factors, it is also observed that there is significant genetic heterogeneity between hormone-sensitive cancers. This finding will have an implication in future research to investigate the complex biological pathways of carcinogenesis that may result in a new opportunity for early detection of hormone-sensitive cancers in precision health.

## Methods

**Study design and cohort characteristics.** We used data in the UK Biobank (UKB) (http://www.ukbiobank.ac.uk). The UKB is a large prospective study that aims to improve the diagnosis, treatment, and prevention of disease. Full details are described elsewhere[65]. It includes more than 500,000 participants aged 37–73 years, with baseline recruitment conducted between 2006 and 2010. Informed consent was obtained during enrolment, as was permission to access medical and other health-related data for research purposes. The UKB has approval from the Northwest Multi-Centre Research Ethics Committee (MREC) and the National Information Governance Board for Health and Social Care (11/NW/0382).

Cancer status was ascertained through linkage to national cancer registries[66]. Information on cancer registration was available up to October 2016, which includes the diagnoses code according to the International Classification of Diseases (ICD; ninth and tenth editions). We mapped all cancer-related ICD codes into "phecodes" which better reflect disease coding as relevant for clinical practice[67]. We excluded participants who had self-reported having had cancer but did not have a record in the cancer registry. For participants with multiple cancer diagnoses, we included the first diagnosed cancer based on the date of diagnosis. As controls, we used participants with no report of any type of cancer-based on self-report, cancer registry, or hospital inpatient data, or benign or in situ tumours from the cancer registry. The World Health Organisation International Agency for Research on Cancer (WHO/IARC) identifies certain cancer sites linked to overweight and obesity as obesity-related cancers[68]. As dysfunctional adiposity is known to associate with hormonal changes, applying criteria previously used by others[9], we classified a subset of obesity-related cancers as "hormone-sensitive cancers" (namely postmenopausal breast, uterine, ovary, prostate, and thyroid). Our analyses included 235,512 controls and 15,197 hormone-sensitive cancer cases. Incident cancer cases were defined as those diagnosed after the baseline assessment and before the end of follow-up (October 2016) and prevalent cases were those diagnosed before baseline assessment in the UKB. So, 7038 incident cases of hormone-sensitive cancer were included. The detailed number of cases of hormone-sensitive cancer is included in Supplementary Table 6. The basic covariates and covariates used for statistical adjustments are described in detail in the supplementary file (Supplementary method).

*Genotypic data.* To control for artifacts introduced to the data during genotyping, initial standard quality control (QC) measures were applied to all data sets before analyses. The genotype data in the UKB includes 92,693,895 SNPs genotyped from 488,377 study participants. The QC procedure for the genotypic data focused on two levels i.e., at individual and SNP level. First, at the individual level, we exclude individuals with a call rate of less than 95% and individuals who did not self-identify as white British ancestry or who exhibited sex inconsistencies (sex mismatch between self-reported phenotype sex and genotype determined sex data) and had a putative sex chromosome aneuploidy (chromosomal anomalies). To check identical genes shared through common ancestors, we randomly selected individuals from a pair and excluded those pairs in which their genomic relationship is larger than 0.05. Furthermore, to avoid bias induced as a result of population stratification and to ensure participants are taken from a relatively homogenous population, we checked the population substructure in the Principal Component (PC) analysis the excluded individual as population outliers with the first or second PC outside ±6 SD of the population mean. Based on the release of the UKB genotype dataset, for those who were included in both the first and second, we calculated the genotype discordance rate between imputed genotype of the two versions for each SNP and each individual and exclude those with a genotype discordance rate of more than 0.05. Secondly at the SNP level, genetic markers with an INFO score <0.6, markers that deviate significantly from Hardy–Weinberg equilibrium (HWE) (1.00E-07) or with a call rate <0.95, with MAF < 0.01 and ambiguous or duplicated SNPs were excluded. Additional specific cohort-level quality control measures can be found in the reference cohort-specific publications[69]. To avoid systematic differences between cases and controls being interpreted as genetic variance, a more stringent quality-control process was then applied to the data. This included excluding individuals with incomplete phenotype data and re-moving markers with a minor allele frequency of less than 1%. In this study, we used high-quality SNPs from the International HapMap Project [HapMap3] that were reliable in estimating genetic variance and covariance at the genome-wide level, feasible for more complicated analyses and there was no

substantial difference between estimated genetic variance from HapMap3 and 1000 genome SNPs[70]. After QC, 1,217,312 HapMap3 SNPs with 288,837 study participants have remained for the analyses.

**Statistical information.** For the Univariate heritability estimate, we assumed a linear mixed model for the heritability analysis as follows:

$$y = Xb + Za + \varepsilon \tag{1}$$

where $y$ is a vector of the response variable (cancer status); $b$ is the vector of regression coefficients for the fixed effects; $a$ is additive genetic effects with variance; $\varepsilon$ is residual (environment effects) with variance and $Z$ and $X$ is the design of matrix of the fixed effects[27].

For the heritability estimate, the genomic relationship matrix (GRM) was constructed using plink software[71,72]. To estimate the Univariate heritability of the subgroups of cancers, two different methods were applied. First, we used the genomic relationship matrix-restricted maximum likelihood (GREML) method, which is based on the individual level genotype data. Second, as linkage disequilibrium score (LDSC) regression method largely depends on summary level genotype data, using the UKB individual genotype data, we computed the summary statistics. We used the pre-computed LD score for white Europeans[73] which is considered suitable for standard LDSC analysis in European populations to use it in a command-line tool of LDSC. For each method, we used both incident and prevalent cases together in the dataset as cases. The analyses were repeated restricting incident cancer cases only. With the use of the prevalence rate of the subgroups of cancers, the observed scale estimates were transformed to liability scale according to Lee et al using MTG2 software. We used $\chi^2$ which is distributed following a chi-square distribution with 2 degrees of freedom and Wald tests.

The GREML method requires individual-level genotype data and is computationally demanding[71]. The sample size of the UKB is large, therefore, we randomly subdivided the dataset to shorten computing time and applied a meta-analysis approach. We first divided the samples into two groups, UKBB1 (91,472 individuals from the first release) and the other samples except for UKBB1, named as UKBB2. In UKBB2, 197,365 individuals with genotype data passed the QC. We further randomly divided the UKBB2 into two groups of equal size (denoted as UKB2A [$n = 98,682$] and UKB2B [$n = 98,683$]) and fitted all models mentioned above for each group. We then meta-analysed the heritability and other related estimates from UKB2A, UKB2B, and UKBB1 using the Fisher's method[74]. For UKBB2, we used the same variables for adjustment as UKBB1.

**Genome-wide association (GWAS) analysis.** Recent advances in computational methods have facilitated the investigation of genetic variants and their effects on multiple complex diseases, i.e., GWAS. After estimating heritability, we, therefore, extend the analysis to estimate the effects of genome-wide SNPs associated with causal genes on the group of hormone-sensitive cancers as a single trait GWAS, using a logistic regression model. The phenotype used for the GWAS analysis is similar to the SNP-based heritability estimate. In total, 15,197 hormone-sensitive cancer cases, including breast cancer, prostate, uterine, ovarian, thyroid, and 223,207 controls were included in the GWAS analysis. The phenotype is similarly adjusted to multiple variables to the heritability estimate to identify significant SNPs using the list of common SNPs from HapMap3. We first computed the statistical power of the study for hormone-sensitive cancers using the online available software GAS Power calculator for genomic study[75]. The power calculation is conducted under the assumptions of genetic models (i.e., additive), 5% minor allele frequencies (MAFs), pair-wise LD, a 6.34% disease prevalence, 1:1 case-to-control ratio, and 5% level of significance. We found the sample size of hormone-sensitive cancers was sufficient to achieve 80% statistical power according to the additive genetic model applied. The power curve is attached in the supplementary file (Supplementary Fig. 6).

We performed post GWAS analyses that involves constructing a quantile-quantile (QQ) plot for hormone-sensitive cancers in each case [all hormone-sensitive cancer cases vs incident hormone-sensitive cancer cases only]. We further quantified the degree of genomic inflation factor ($\lambda$) i.e., how best the observed data points fit to the expected value. The QQ plots in each case showed the bulk of the distribution is in the lower tail of the graph.

We identified genome-wide significant SNPs for hormone-sensitive cancers using plink software[72] to obtain the GWAS $P$-values that were used for the Manhattan plot for *qqman* package in R. For the post GWAS analysis to see if the genomic inflation factor is high, we plot QQ plot using QCEWAS package in R. $\lambda$ is the median of the resulting chi-square test statistics divided by the expected median of the chi-square distribution. The median of a chi-squared distribution with one degree of freedom is 0.4549364, i.e., [qchisq (0.5,1) = 0.4549364]. A $\lambda$ value is calculated from p-values in the output we have from the genome-wide association analysis. Low significant results are removed (there are more significant results than expected) to increase the $\lambda$ value. To rescale the $\lambda$ value to provide better information, we use the following formula to rescale the $\lambda$ calculated[76].

$$\lambda_{1000} = 1 + (\lambda\, observed - 1) \times \frac{\left(\frac{1}{n\,cases} + \frac{1}{n\,controls}\right)}{\left(\frac{1}{n\,cases,1000} + \frac{1}{n\,controls,1000}\right)}, \tag{2}$$

where $n$ is the study sample size for cases and controls respectively, and ncases,1000 and ncontrols, 1000 is the target sample size of 1000.

**Phenotypic correlation**. Estimates of phenotypic and genetic correlation were computed separately between hormone-sensitive cancer and each non-cancer trait. The phenotypic correlation was estimated using Pearson correlations between each pair of traits for complete observation in R. To examine the genetic architecture further, we performed phenotypic correlation for components of hormone-sensitive cancers using the leave-one-out analysis approach. The results are summarised and presented in Table 3.

**Genetic correlation analysis**. As bivariate LDSC estimates are not biased with sample overlap wherein controls are common in both traits and computationally very efficient[37], we run the genetic correlation to generate an overview of the genetic relationship between hormone-sensitive cancers and the six non-cancer subgroup traits. We then used the bivariate GREML approach to estimate the genetic correlation between hormone-sensitive cancers and seven non-cancer traits. Further, we examine the genetic correlation between each component of hormone-sensitive cancers using a pair-wise comparison approach. The genetic correlation ($r_g \pm SE$) is calculated using cross-trait LD Score regression method.

As most oestradiol hormone is bound to the serum protein sex-hormone binding globulin (SHBG) and Albumin, i.e., biologically unavailable to exhibit its physiologic effect, implying the need to compute the free hormone level, we calculated the free concentration using serum oestradiol and the concentration of SHBG and Albumin with their respective association constant K[77].

$$cFO = \left(E_2 - N_{Total}\right)/\left(\left(N_{SHBG}\right) - \left(E_2 + N_{E_2}\right)\right) \qquad (3)$$

where cFo = calculated free oestradiol; $E_2$ = serum oestradiol level; $N_{E2} = 0.64 \times 10^9$*Albumin level +1; $N_{SHBG} = 5.55 \times 10^4$ *SHBG level; and $N_{TOTAL} = N_{SHBG} + N_{E2}$.

**Leave-one-out (LOO) approach to determine the genetic correlation of hormone-sensitive cancers**. The iterative scheme of leave-one-out analysis is carried out by using a different possible combination of hormone-sensitive cancers in cross-trait LDSC regression. The grouped hormone-sensitive cancer comprised of five distinct heterogeneous cancers, and we created a 5-fold leave-one-out analysis that involves the different possible combinations of the hormone-sensitive cancers. During each iterative step, we exclude data of one independent cancer at a time and use the remaining cancer types as grouped hormone-sensitive cancers to compute the genetic correlation in bivariate LDSC. These steps are iteratively completed five times. The analysis sketch map demonstrating all the possible combinations is summarised in Supplementary Fig. 5.

**Gene-environment interaction**. Finally, we checked the gene-environment interaction for hormone-sensitive cancers with selected traits using bivariate GREML and GxEsum techniques for traits with continuous level measurement. The bivariate GREML approach is applied with the assumptions of gene-environment interactions in contrast to the Univariate GREML model that assumes the absence of GxE interactions. Here in this method, we stratified the hormone-sensitive cancer phenotype by traits regarded as environments [i.e., BMI-normal vs high; metabolic environment-favourable vs unfavourable; and sex-male vs female] to look for interactions. Such approach allows us to test whether the genetic effects are heterogeneous if individuals lie in the same environment thereby test for gene-environment interaction.

A recently proposed alternative method for quantitative traits, called GxEsum is able to estimate gene-environment interaction. This method is built on LDSC approach by using GWAS summary statistics and suggested as computationally efficient method[39]. For SNP effects modulated by quantitative environment, the expected chi-square statistics ($\chi_j^2$) is

$$E\left[\chi_j^2 | \ell_j\right] = \frac{N\sigma_{g_1}^2}{M} * \ell_j + 1 + 2\left(\sigma_{g_1}^2 + \sigma_{\tau_1}^2\right), \qquad (4)$$

where $N$ is the number of individuals, $M$ is the number of SNPs, $\sigma_{g_1}^2$ is the variance due to GxE, $\sigma_{\tau_1}^2$ is the variance due to residual heterogeneity or scale effects caused by residual-environment interaction (RxE), $\ell_j$ is the LD score at the variant j.

*Software*. We have used the well-established MTG2[27] software to conduct the bivariate GREML analyses and estimate the genetic correlation coefficient between each non-cancer trait and subgroups of cancer. For MTG2, the source code, executive binary file, user manual, and toy examples for practice are readily available for downloads using the link https://sites.google.com/sit/honglee0707/mtg2. The GxEsum model is implemented in the script that is publicly available at https://github.com/honglee0707/GxEsum. The version of source code used in the manuscript is deposited with https://doi.org/10.5281/zenodo.4659681 at https://zendo.org/record/4659681#.YGKZXc9xeUk. The rest statistical analyses were performed using publicly available software that includes plink1.9, LDSC, and analysis packages in R & Python.

*Statistics and reproducibility*. We used an appropriate linear mixed model of GREML, and logistic regression based GWAS models as described in each section of the manuscript. The $P$-value in this study is calculated by applying the Wald-test with the assumption of the distribution of estimated genetic correlation was normal. The statistical significance level was set at $p < 0.05$ (2-tailed). We confirmed the reproducibility of the main analyses by randomly splitting the UKB data into two datasets as UKB1 and UKB2 fitted all statistical models.

**Reporting summary**. Further information on research design is available in the Nature Research Reporting Summary linked to this article.

## Data availability

All data will be available to approved users of the UK Biobank upon application. The data are not publicly available due to the contents related to information that could compromise research participants' privacy/consent. The authors state that all data necessary for confirming the conclusions presented in the manuscript are represented fully within the manuscript. Source data is provided as Supplementary Data 1. Individual-level genotype data are available by application to the UK Biobank. The GWAS summary statistics dataset that is generated during the current study and supports the findings have been deposited in the NHGRI-EBI GWAS catalogue with the accession codes GCST90102435, GCST90102436; GCST90102437; GCST90102438; GCST90102439; GCST90102440; GCST90102441; (https://www.ebi.ac.uk/gwas/).

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

## Acknowledgements

The authors extend sincere thanks to the study participants of the UK biobank, who made this work possible. This study has been conducted using the UK Biobank Resource (http://www.ukbiobank.ac.uk) with the reference number 14575 & 20175 approved by

UK Biobank. The High-Performance Computing (HPC) resources were provided by the Australian government through Gadi under the National Computational Merit Allocation Scheme (NCMAS) and by the University of South Australia through super-computing clusters of Tango 2.0 and StatGen server. This work is supported by grants from the Australian Research Council (DP 190100766); Tour de Cure (RSP-013-18/19) and National Health and Medical Research Council, Australia (GT1157281).

## Author contributions

M.A. wrote the manuscript, conducted data management and statistical analysis. S.H.L. developed concept and designed the study; S.H.L. and E.H. jointly supervised the work and funded the study; V.-P.M., A.M., T.B., and J.S. provide critical input on the result interpretation and edited the manuscript. All authors interpreted the data, critically reviewed the manuscript and approved the final version for submission.

## Competing interests

The authors declare no competing interests.
