## [Peer Review File · Communications Biology]

Reviewers' comments:

Reviewer #1 (Remarks to the Author):

Overall an interesting analysis of hormone-sensitive cancers, to evaluate for common genetic factors in carcinogenesis of these tumors, compared with non-hormone sensitive cancers.

Many cancers share some genetic mechanisms in common, but the authors are hypothesizing that hormone-sensitive carcinogenesis per se has a unique set of common factors. However, they don't provide a clear rationale for this. For example, why would an estrogen-sensitive cancer have factors in common with an androgen-sensitive cancer, or a TSH-sensitive cancer? Do they have common target genes? or common transcriptional co-factors? etc. (in other words, why should they be considered, in this analysis, as a single disease?). There are potentially good reasons for hypothesizing this, but the authors don't develop this idea, either in the introduction or discussion; and they don't relate their findings to that central question; for example, do the genes covered by the 55 SNP's have known interactions between different hormone-sensitive signaling pathways, but not (presumably) signaling pathways involved in non-hormone-sensitive cancers? Why would FGF2, or POU5F1B, etc., be links specifically between hormone-sensitive cancers?

Overall, the statistical analyses are valid.

A few other issues:

- it is not clear to this reviewer that all their "hormone-sensitive cancers" were actually hormone sensitive; for example, were triple-negative breast cancers separated out from ER/PR+ breast cancers?
- although the methods and rationale of the heritability estimates, and GREML analysis, are very well described in the Supplementary Note, it would be helpful to incorporate some of this into the Results section, when introduced, to clarify why these are being done; for example, something like "GREML is a statistical method that estimates the amount of variance in one or more phenotypes attributable to a collection of genetic polymorphisms; therefore, we applied it to address ...", instead of just launching into its use in the Results section (this journal is not a statistical genetics journal, but rather has a broad, general audience, that will likely not have expertise in these methods).

In summary, this work is very interesting and promising, but requires further development before I could recommend it for publication in this journal, and should be submitted to a journal more focused on cancer or statistical genetics.

Reviewer #2 (Remarks to the Author):

This is an interesting manuscript that describes findings from evaluating 1) the genetic basis of hormone-sensitive cancers by treating five cancers that are sensitive to hormone levels (breast, uterine, ovarian, prostate, and thyroid cancer) as a single disease and 2) the genetic relationship of hormone-sensitive cancers with other cancer types and cancer-related factors. The authors evaluated the heritability of this overall hormone-sensitive cancers and compared it to the heritability of overall cancer, obesity-related cancer, and finally the site-specific hormone-sensitive cancers from previous studies. A GWAS of this hormone-sensitive cancers phenotype was performed and 55 genome-wide significant SNPs were reported. Phenotypic and genetic correlations were calculated between hormone-sensitive cancers and cancer-related factors. Pairwise genetic correlations between breast, prostate, uterine, and three other cancer types were also reported. Genetic correlations calculated using a leave-one-out approach were reported to show the genetic relationship of each of the five hormone-sensitive cancers with the rest of the cancers and three other cancer types. Finally, gene-environment interaction analyses were performed by treating the hormone-sensitive cancers as the phenotype and sex, BMI, and metabolic environment as environmental variables.

This is a comprehensive study of the genetics of hormone-sensitive cancers. The findings provide

insights into the shared genetics of hormone-sensitive cancers at both the genome-wide and SNP levels. One strength is that the analyses were performed on all cases and incident cases separately; comparing the results from these two sets of analysis may tell us more about the mechanism underlying the observed genetic correlation. However, I suggest the authors further improve the methods as well as the overall writing and structure of the main text, tables, and figures to strengthen the validity of the study.

Some specific points:

1. Maybe I missed this, but the number of cases for each hormone-sensitive cancers was not provided in the main text or tables. If treating all hormone-sensitive cancers as a single disease (if I understand this correctly, subjects diagnosed with any hormone-sensitive cancer are cases, cancer-free subjects are controls, and GWAS was performed on this new phenotype), I would expect the relative proportion of cancer cases for each cancer type to have a substantial influence on the results. One extreme case is that if all cases are breast cancer cases, the GWAS results would not reflect any underlying genetic variations of other cancers (i.e., not reflecting common effects *shared* by all cancers). If the case numbers are small for some cancer types, e.g., ovarian and thyroid, then the genetics of those cancer types may not be reflected by studying this overall phenotype.

2. Instead of treating the five hormone-sensitive cancers as a single disease, it might be better to perform a meta-analysis of the single-trait GWAS (e.g., using MTAG) to identify the shared genetic variations across these hormone-sensitive cancers and reveal any shared mechanism. In addition to listing the identified SNPs in the supplementary table, it would be helpful to show the identified independent loci in a main table, along with their effects on each of the hormone-sensitive cancers.

3. Line 71-72: When treating multiple hormone-sensitive cancers as a single disease, estimated SNP-based heritability can inform if the common germline variants contribute to the carcinogenic risk shared between hormone-sensitive cancers.

Again, I think the heritability of this single disease may not capture the shared genetics across these hormone-sensitive cancers very well. It reflects the genetic variation of being diagnosed with any of the hormone-sensitive cancers, but not necessarily shared by these cancers. Instead, the genetics correlations between these cancers may reflect the shared genetics at the genome-wide scale; shared susceptibility loci may reflect common mechanisms.

4. Table 3 doesn't include thyroid and ovarian cancer, which seems like a sample size issue as the authors described in the leave-one-out analysis. But it is not clear that why the other three cancer types were selected and included (i.e., why select these cancer types but not others and what is the motivation of including other cancer types). Please clarify in the main text.

Some other comments:

1. The title of this manuscript sounds like it is describing the heritability of hormone-sensitive cancers and the genetic correlations between these cancers. However, it seems like the main focus of the main text is the genetic correlations of these hormone-sensitive cancers with other cancer types and cancer-related factors, but not between these cancers (at least it was not estimated for all hormone-sensitive cancers). Also, genetic correlation was not mentioned in the abstract.

2. The use of prospective vs. retrospective and incidence vs. prevalence cases is mixed in the text and tables. These terms should be defined clearly at the beginning of results and the usage should be consistent in text and tables.

3. "Female specific factors [age at menopause]" was mentioned in line 151, but Table 2 reports "Women Factors [Menopausal Status]". Please clarify. It's better to name this category as "Menstrual and reproductive factors" or just "Menstrual factors".

4. Table 4 is difficult to read – it might be better to replace 'Hormone-sensitive-X' with 'Excluding

XXX' or something else that is easy to understand with only the information in the table.

5. In methods, the five selected hormone-sensitive cancers are cancers of the breast, endometrium, ovary, prostate, and thyroid, but in other parts of the manuscript, it seems that 'uterine cancer' was used to refer to 'endometrial cancer', please be consistent.

6. Figure 3 and 4. It seems that the values are percentages but there is no annotation.

Responses to Reviewers' comments

Reviewer #1

(Remarks to the Author):

1. Overall, an interesting analysis of hormone-sensitive cancers, to evaluate for common genetic factors in carcinogenesis of these tumors, compared with non-hormone sensitive cancers.

Many cancers share some genetic mechanisms in common, but the authors are hypothesizing that hormone-sensitive carcinogenesis per se has a unique set of common factors. However, they don't provide a clear rationale for this. For example, why would an estrogen-sensitive cancer have factors in common with an androgen-sensitive cancer, or a TSH-sensitive cancer? Do they have common target genes? or common transcriptional co-factors? etc. (in other words, why should they be considered, in this analysis, as a single disease?). There are potentially good reasons for hypothesizing this, but the authors don't develop this idea, either in the introduction or discussion; and they don't relate their findings to that central question; for example, do the genes covered by the 55 SNP's have known interactions between different hormone-sensitive signaling pathways, but not (presumably) signaling pathways involved in non-hormone-sensitive cancers? Why would FGF2, or POU5F1B, etc., be links specifically between hormone-sensitive cancers?

Authors' response: We thank the reviewer for this excellent suggestion. As suggested by the reviewer, we have included the rationale for combining five types of hormone-sensitive cancers as single disease, considering the common target genes and transcriptional cofactors, in the introduction section of the revised manuscript (lines 61-72).

The revised content now reads as:

"The role of common target genes and transcriptional cofactor has received a considerable attention in the biology of hormone-sensitive cancers. For example, the expression of fibroblast growth factor (FGF-2) gene is a signaling molecule with fundamental roles in tumor growth and progression associated with breast, ovarian, thyroid, prostate, and uterine cancers^{10,11,12,13,14}. Furthermore, Michailidou et al., (2013) reported that multiple genomic regions flanking common target genes, such as telomerase reverse transcriptase (TERT) and the POU domain class 5 transcription factor 1B (POU5F1B), included susceptibility loci that were common to breast, prostate, and ovarian cancers, supporting the hypothesis of a common genetic etiology among these cancer types^{15,16,17,18}."

In addition, we have provided a clear rationale and related our findings to the central question as (lines 69-72):

“This growing evidence for the role of common gene signaling in tumorigenesis leads to the proposal of a combined analysis of multiple hormone-sensitive cancers (e.g., treating them as a single disease) to investigate the common genetic aetiology and identify new risk loci underlying the common pathway. ”

In the discussions section, we also have added a sentence that reflect this (lines 351-360), i.e.,

“Moreover, our hypothesis of common shared aetiology is supported by the finding that multiple SNPs in the FGF-2 and POU5F1B gene region are associated with hormone-sensitive cancers, implying the presence of commonly expressed genomic regions. Evidence suggests that gene overexpression may lead to increased angiogenesis and autocrine stimulation of cancer cells ⁴³. For example, there is strong evidence that FGF-2 ligands and receptors are important in breast cancer tumorigenesis ¹⁰. POU5F1B has been suggested to be involved in prostate cancer pathogenesis ¹⁶. It has been demonstrated that hormones can express FGF-2 and POU5F1B genes and the level is progressively elevated during initial phases of tumorigenesis, and further its expression is higher in cancerous tissue in comparison with adjacent normal tissue or benign ones ^{12, 14}”.

2. Overall, the statistical analyses are valid.

Authors’ response: Thank you for your positive assessment of our statistical analyses.

A few other issues:

3. It is not clear to this reviewer that all their "hormone-sensitive cancers" were actually hormone sensitive; for example, were triple-negative breast cancers separated out from ER/PR+ breast cancers?

Authors’ response: We thank the reviewer for this valid point. We could not further classify pathologic-based subtypes of breast cancer, including estrogen-receptor [ER] positive/negative, progesterone-receptor [PR] positive/negative, and triple-negative because the information on these subtypes of breast cancer was not available to us.

Triple-negative breast cancer (TNBC) is characterized by the negative test to estrogen and progesterone receptors, and excess human epidermal growth factor receptor 2 (HER2) protein that accounts for approximately 15% of breast cancers diagnosed worldwide ¹, implying that TNBC is a relatively rare subtype. Furthermore, TNBC is more commonly diagnosed in women younger than 40 years ², and the breast cancer cases included in our analyses are mainly cases of postmenopausal breast cancer, which is an obesity-related cancer [mentioned in the method section lines 464-468, page 19 as hormone-sensitive cancer cases are subset of obesity-related cancer cases namely postmenopausal breast, uterine, ovary, prostate, and thyroid"]. Therefore, it is likely that TNBC cases are negligible in our analyses

so have had little to influence on our findings. Nonetheless, we agree that this is a limitation of our study and explicitly discuss in the text (lines 435-437). The limitation stated as:

“Further, we did not classify subtypes of breast cancer, such as estrogen receptor [ER] positive/negative, progesterone receptor [PR] positive/negative, and triple-negative breast cancers, in our analyses because no such information was available to us.”

4. Although the methods and rationale of the heritability estimates, and GREML analysis, are very well described in the Supplementary Note, it would be helpful to incorporate some of this into the Results section, when introduced, to clarify why these are being done; for example, something like "GREML is a statistical method that estimates the amount of variance in one or more phenotypes attributable to a collection of genetic polymorphisms; therefore, we applied it to address ...", instead of just launching into its use in the Results section (this journal is not a statistical genetics journal, but rather has a broad, general audience, that will likely not have expertise in these methods).

Authors' response: Thank you for this helpful suggestion. The details for the method description were given in the supplementary notes to shorten the main manuscript, however, as per the reviewer's suggestion, we have now added further detail in the result section of the revised manuscript [lines 101-104].

The revised section now reads as:

“Here we used a Genomic Restricted Maximum Likelihood (GREML) analysis, which is a statistical method that estimates the proportion of variance on one or more phenotypes attributed by all genetic polymorphisms using individual-level data to estimate the variance explained by all genetic polymorphisms (SNP-based heritability)”.

5. In summary, this work is very interesting and promising, but requires further development before I could recommend it for publication in this journal, and should be submitted to a journal more focused on cancer or statistical genetics

Authors' response: Thank you for your constructive review. As the reviewer commented that this work is promising, and we have explicitly addressed the reviewer's concerns on the previous version of the manuscript, we hope the manuscript is now acceptable.

Reviewer #2

(Remarks to the Author):

1. This is an interesting manuscript that describes findings from evaluating 1) the genetic basis of hormone-sensitive cancers by treating five cancers that are sensitive to hormone levels (breast, uterine, ovarian, prostate, and thyroid cancer) as a single disease and 2) the genetic relationship of hormone-sensitive cancers with other cancer types and cancer-related factors. The authors evaluated the heritability of this overall hormone-sensitive cancers and compared it to the heritability of overall cancer, obesity-related cancer, and finally the site-specific hormone-sensitive cancers from previous studies. A GWAS of this hormone-sensitive cancer phenotype was performed and 55 genome-wide significant SNPs were reported. Phenotypic and genetic correlations were calculated between hormone-sensitive cancers and cancer-related factors. Pairwise genetic correlations between breast, prostate, uterine, and three other cancer types were also reported. Genetic correlations calculated using a leave-one-out approach were reported to show the genetic relationship of each of the five hormone-sensitive cancers with the rest of the cancers and three other cancer types. Finally, gene-environment interaction analyses were performed by treating the hormone-sensitive cancers as the phenotype and sex, BMI, and metabolic environment as environmental variables. This is a comprehensive study of the genetics of hormone-sensitive cancers. The findings provide insights into the shared genetics of hormone-sensitive cancers at both the genome-wide and SNP levels. One strength is that the analyses were performed on all cases and incident cases separately; comparing the results from these two sets of analysis may tell us more about the mechanism underlying the observed genetic correlation. However, I suggest the authors further improve the methods as well as the overall writing and structure of the main text, tables, and figures to strengthen the validity of the study.

Authors' response: We thank the reviewer for this summary and positive comments. The tabular presentation has been improved, and we have added annotations for tables and figures if needed (see point-by-point responses below).

Some specific points:

2. Maybe I missed this, but the number of cases for each hormone-sensitive cancer was not provided in the main text or tables. If treating all hormone-sensitive cancers as a single disease (if I understand this correctly, subjects diagnosed with any hormone-sensitive cancer are cases, cancer-free subjects are controls, and GWAS was performed on this new phenotype), I would expect the relative proportion of cancer cases for each cancer type to have a substantial influence on the results. One extreme case is that if all cases are breast cancer cases, the GWAS results would not reflect any underlying genetic variations of

other cancers (i.e., not reflecting common effects *shared* by all cancers). If the case numbers are small for some cancer types, e.g., ovarian and thyroid, then the genetics of those cancer types may not be reflected by studying this overall phenotype.

Authors' response: We appreciate the reviewer for this suggestion. We have now incorporated the tabulated information and revised sentences in the main text about the number of cases for each hormone-sensitive cancer for more clarity. Please see the revised manuscript (lines 471-473) and Supplementary Table 6.

The revised sentence in the main text now reads as:

“So, 7,038 incident cases of hormone-sensitive cancer were included. The detailed number of cases of hormone-sensitive cancer is included in Supplementary Table 6”.

3. Instead of treating the five hormone-sensitive cancers as a single disease, it might be better to perform a meta-analysis of the single-trait GWAS (e.g., using MTAG) to identify the shared genetic variations across these hormone-sensitive cancers and reveal any shared mechanism. In addition to listing the identified SNPs in the supplementary table, it would be helpful to show the identified independent loci in a main table, along with their effects on each of the hormone-sensitive cancers.

Authors' response: We thank the reviewer for this suggestion. We have now added the results of the meta-analysis from the 5 single-trait GWASs using the command `--meta-analysis` in PLINK 1.9 (a fixed-effect inverse-variance weighted method) (Supplementary Table 10 and Supplementary Fig. 4). From the results, we observed that 37 genome-wide significant SNPs were identified at chromosome 9 only, indicating that the meta-analysis is less powered, compared to the analysis of combined hormone-sensitive cancers as a single disease (that identified 55 genome-wide significant SNPs in chromosome 2, 8, 10, 11, 16, 17, and 19). When carrying out the meta-analysis of the single-trait GWAS for the incident cases only, no genome-wide significant SNPs were found, noting that the analysis of combined hormone-sensitive cancers as a single disease for incident cases only identified 33 genome-wide significant SNPs in chromosome 8, 10, 11, and 17.

Our analysis supports the conceptual premise of the combined analysis that can capture common genetic risk factors shared between hormone sensitive cancers, some of which may not be identified by the meta-analysis of single-trait GWASs. This is presented in the result section (lines 136-138 and Fig.2) in comparison with the meta-analyzed finding presented in lines 161-169, and in Supplementary Fig. 4.

When considering LD, the number of independent loci ($LD\ r^2 > 0.2$) is 12 SNPs (Table 2) for the identified 55 SNPs from the analysis of combined hormone-sensitive cancers as a single disease (a plot showing the LD heatmap in Supplementary Fig. 2a). For the analysis restricted

to incident hormone-sensitive cancer cases, we identified 8 genome-wide significant independent loci for the identified 33 SNPs (Supplementary Table 9) and the LD heatmap in Supplementary Fig. 2b.

4. Line 71-72: When treating multiple hormone-sensitive cancers as a single disease, estimated SNP-based heritability can inform if the common germline variants contribute to the carcinogenic risk shared between hormone-sensitive cancers.

Again, I think the heritability of this single disease may not capture the shared genetics across these hormone-sensitive cancers very well. It reflects the genetic variation of being diagnosed with any of the hormone-sensitive cancers, but not necessarily shared by these cancers. Instead, the genetic correlations between these cancers may reflect the shared genetics at the genome-wide scale; shared susceptibility loci may reflect common mechanisms.

Authors' response: We explicitly estimated genetic correlations between breast vs. prostate ($r_g = 0.10$, $SE=0.09$), breast vs. uterine ($r_g = 0.32$, $SE= 0.20$), and prostate vs. uterine cancers ($r_g = 0.12$, $SE=0.18$) (Fig. 5). We also estimated genetic correlations between breast cancer vs. hormonal cancer excluding breast cancer ($r_g = 0.1662$, $SE= 0.0930$) and prostate cancer and hormonal cancer excluding prostate cancer ($r_g = 0.2209$, $SE=0.1101$) (Table 4). These estimates suggest that there is significant genetic heterogeneity among these cancers. However, we also would like to quantify how much phenotypic variance is explained by the common genetic factors shared among hormone-sensitive cancers, which can be assessed by SNP-based heritability on the overall hormone-sensitive cancer coded as a single disease.

We have clarified this in the revised manuscript (lines 268-271).

“While these estimates suggest that there is significant genetic heterogeneity among these cancers, the estimate of SNP-based heritability of the overall hormone-sensitive cancer coded as a single disease shows that the phenotypic variance explained by the common genetic factors is significantly different from zero (Fig. 1).”

5. Table 3 doesn't include thyroid and ovarian cancer, which seems like a sample size issue as the authors described in the leave-one-out analysis. But it is not clear that why the other three cancer types were selected and included (i.e., why select these cancer types but not others and what is the motivation of including other cancer types). Please clarify in the main text.

Authors' response: Thank you for bringing attention to the clarity needed in the cancer definition for hormone-sensitive cancers. We have now clarified that hormone-sensitive cancers are a subset of the obesity-related cancers identified by the World Health Organization (WHO) International Agency for Research on Cancer (IARC) ³. We then

grouped the five cancers that share a characteristic mechanism of carcinogenesis that involves hormones from the list of obesity-related cancer by WHO/IARC (lines 464-468, page 19). Those additional cancers included in the genetic correlation analysis i.e., colorectal, renal, and multiple myeloma are from these obesity-related cancers with the intent of looking for genetic correlation as there is sex difference in the incidence of these cancers implying that the association between sex hormones and genetic variants in hormone metabolic pathways might have a role^{4,5,6}. We have now added a sentence to clarify the need for including other obesity-related cancer in the genetic correlation analysis of the leave-one-cancer-out analysis (lines 254-256).

"We further carried out genetic correlation analyses into grouped hormone-sensitive and other obesity-related non-hormone sensitive cancers in the UKB (namely colorectal, kidney, and multiple myeloma to gain more detailed understanding of the complexities of hormone-cancer phenomena".

We also clarified as to why ovarian and thyroid cancers were not included in the footnotes of Fig. 5 and Table 4. For example, the footnotes in Fig. 5 reads as:

"Ovarian and thyroid cancers were not estimable, which was probably due to the fact that the number of cases was not sufficient for LDSC in the analysis of these diseases."

Some other comments:

6. The title of this manuscript sounds like it is describing the heritability of hormone-sensitive cancers and the genetic correlations between these cancers. However, it seems like the main focus of the main text is the genetic correlations of these hormone-sensitive cancers with other cancer types and cancer-related factors, but not between these cancers (at least it was not estimated for all hormone-sensitive cancers). Also, genetic correlation was not mentioned in the abstract.

Authors' response: Thank you for this insightful suggestion. The title has been modified to reflect the main focus on the analysis of hormone-sensitive cancers as a single disease. The revised title now reads as:

"Heritability of Hormone-Sensitive Cancers as a single disease in the UK Biobank: A molecular Evidence of Shared Aetiology"

In addition, we made it clear that we estimated the genetic correlation between each pair of hormone-sensitive cancers except the pairs that were not estimable (lines 236-240, page 10, Figure 5, and Table 4). Please also see responses to Q4 and Q5 above). We have now revised the abstract as (lines 45-47)

"Pair-wise analysis also estimated positive genetic correlation between some pairs of hormone-sensitive cancers although they were not statistically significant".

7. The use of prospective vs. retrospective and incidence vs. prevalence cases is mixed in the text and tables. These terms should be defined clearly at the beginning of results and the usage should be consistent in text and tables.

Authors' response: Thank you for noticing this; we have now revised, and the terms are defined clearly, and used consistently with extensive efforts with the aim of improving the clarity of our manuscript.

The terms defined in the method (lines 469-473) reads as:

“Incident cancer cases were defined as those diagnosed after the baseline assessment and before the end of follow-up (October 2016,) and prevalent cases were those diagnosed before baseline assessment in the UKB. So, 7,038 incident cases of hormone-sensitive cancer were included. The detailed number of cases of hormone-sensitive cancer is included in Supplementary Table 6.”

8. “Female specific factors [age at menopause]” was mentioned in line 151, but Table 2 reports “Women Factors [Menopausal Status]”. Please clarify. It’s better to name this category as “Menstrual and reproductive factors” or just “Menstrual factors”.

Authors' response: Thank you for this helpful correction. Required changes have been made in Table 3 and the entire main text. Please see the result section [lines 176-177] and Table 3 of the revised manuscript.

9. Table 4 is difficult to read – it might be better to replace ‘Hormone-sensitive-X’ with ‘Excluding XXX’ or something else that is easy to understand with only the information in the table.

Authors' response: We appreciate this suggestion. We have now updated Table 4 accordingly.

10. In methods, the five selected hormone-sensitive cancers are cancers of the breast, endometrium, ovary, prostate, and thyroid, but in other parts of the manuscript, it seems that ‘uterine cancer’ was used to refer to ‘endometrial cancer’, please be consistent.

Authors' response: Thank you for noticing this. The list of cancers included in the analysis are obesity-related cancers reported by WHO/IARC³, and to be consistent in the cancer types with the agency’s list of cancer, we have now consistently used the term “uterine cancer” throughout the manuscript.

11. Figure 3 and 4. It seems that the values are percentages but there is no annotation.

Authors' response: Thank you for pointing this out. We now have revised the figures to improve clarity. Similarly, corresponding updates have now been included in the footnotes of Fig. 3 and Fig. 4 and in the main text to show the values are percentages of the estimates.

References

1. Rakha EA, El-Sayed ME, Green AR, Lee AH, Robertson JF, Ellis IO. Prognostic markers in triple-negative breast cancer. *Cancer* **109**, 25-32 (2007).
2. Trivers KF, *et al.* The epidemiology of triple-negative breast cancer, including race. *Cancer Causes Control* **20**, 1071-1082 (2009).
3. IARC. Latest global cancer data *World Health Organization Press Release* **263**, (2018).
4. Li S, *et al.* Sex hormones and genetic variants in hormone metabolic pathways associated with the risk of colorectal cancer. *Environ Int* **137**, 105543 (2020).
5. Sun L, *et al.* Impact of Estrogen on the Relationship Between Obesity and Renal Cell Carcinoma Risk in Women. *EBioMedicine* **34**, 108-112 (2018).
6. Hosgood HD, Gunter MJ, Murphy N, Rohan TE, Strickler HD. The Relation of Obesity-Related Hormonal and Cytokine Levels With Multiple Myeloma and Non-Hodgkin Lymphoma. *Front Oncol* **8**, 103 (2018).

REVIEWERS' COMMENTS:

Reviewer #1 (Remarks to the Author):

The authors have done a reasonable job in addressing this reviewer's comments.

I still suggest that they could have been more thorough in the developing their hypothesis and analysis of their results. For example, still not addressed, in developing their hypothesis, is why cancers sensitive to hormones (estrogen, androgen) acting via nuclear hormone receptors should be grouped with cancers sensitive to hormones (TSH) acting via G-protein coupled receptors.

I apologize for a typo in my initial review, which mentioned FGF2, but should be FGFR2; however, the issue remains that FGFR2 (as an example; 8/55 genome-wide significant SNPs in UKB) is definitely implicated in cancers other than their group of hormone-sensitive cancers, and so why do they think inherited variants in this gene are associated with hormone-sensitive cancers.

Reviewer #2 (Remarks to the Author):

Thank the authors for the response to the reviewer's comments.

REVIEWERS' COMMENTS:

Reviewer #1 (Remarks to the Author):

1. The authors have done a reasonable job in addressing this reviewer's comments. I still suggest that they could have been more thorough in the developing their hypothesis and analysis of their results. For example, still not addressed, in developing their hypothesis, is why cancers sensitive to hormones (estrogen, androgen) acting via nuclear hormone receptors should be grouped with cancers sensitive to hormones (TSH) acting via G-protein coupled receptors.

Authors response: We thanks the reviewer for this suggestion and now added the suggested content in formulating our hypothesis in this final version of the manuscript (lines 69-76)

The revised section now reads as:

“Cancers sensitive to hormones also involve the activation of G protein-coupled receptor and nuclear-mediated receptors that triggers multiple cellular signaling events to cause the disease. For example, the G protein-coupled estrogen receptor (GPER) plays an important role in cancer of both male and female reproductive systems¹⁹. Studies also highlighted the binding of nuclear receptors to their respective DNA target motifs across the genome, playing critical roles in the development and progression of cancer^{20, 21, 22}. This growing evidence for the role of common genes and involvement of nuclear mediated and transmembrane signaling in tumorigenesis leads to the proposal of a combined analysis of multiple hormone-sensitive cancers ...

2. I apologize for a typo in my initial review, which mentioned FGF2, but should be FGFR2; however, the issue remains that FGFR2 (as an example; 8/55 genome-wide significant SNPs in UKB) is definitely implicated in cancers other than their group of hormone-sensitive cancers, and so why do they think inherited variants in this gene are associated with hormone-sensitive cancers.

Authors response: Thank you for bringing attention to the clarity needed in the inherited variants associated with hormone-sensitive cancers. FGF2 acts as a ligand for FGFR1 and FGFR2¹ to drive crucial tumorigenesis signaling pathways, which are responsible to cause several cancer types including hormone-sensitive cancers². Some of the genome-wide significant SNPs associated with FGFR2 (Supplementary Table 7) have been also reported to be associated with hormone-sensitive cancers in previous studies, mainly for breast and prostate cancers^{3, 4, 5}. We added “FGFR2 receptors” in line 358.

For example, there is strong evidence that FGF-2 ligand and FGFR2 receptors play an important role in the development and progression of breast and prostate cancer^{6, 7, 8}.

Reviewer #2 (Remarks to the Author):

1. Thank the authors for the response to the reviewer's comment

Authors response: We thank the reviewer for accepting our responses.

References

1. Ornitz DM, *et al.* Receptor specificity of the fibroblast growth factor family. *J Biol Chem* **271**, 15292-15297 (1996).
2. Turner N, Grose R. Fibroblast growth factor signalling: from development to cancer. *Nat Rev Cancer* **10**, 116-129 (2010).
3. Wein L, *et al.* FGFR2 amplification in metastatic hormone-positive breast cancer and response to an mTOR inhibitor. *Ann Oncol* **28**, 2025-2027 (2017).
4. Teishima J, *et al.* Fibroblast Growth Factor Family in the Progression of Prostate Cancer. *J Clin Med* **8**, (2019).
5. Lee JE, Shin SH, Shin HW, Chun YS, Park JW. Nuclear FGFR2 negatively regulates hypoxia-induced cell invasion in prostate cancer by interacting with HIF-1 and HIF-2. *Sci Rep* **9**, 3480 (2019).
6. Rashkin SR, *et al.* Pan-cancer study detects genetic risk variants and shared genetic basis in two large cohorts. *Nat Commun* **11**, 4423 (2020).
7. Jiang L, *et al.* A resource-efficient tool for mixed model association analysis of large-scale data. *Nat Genet* **51**, 1749-1755 (2019).
8. Morra A, *et al.* Association of germline genetic variants with breast cancer-specific survival in patient subgroups defined by clinic-pathological variables related to tumor biology and type of systemic treatment. *Breast Cancer Res* **23**, 86 (2021).